# Structured Neural-PI Control with End-to-End Stability and Output Tracking Guarantees

**Wenqi Cui**[1]  **Yan Jiang**[1]  **Baosen Zhang**[1]  **Yuanyuan Shi**[2]

[1]University of Washington, WA 98195   [2]University of California San Diego, CA 92093

`wenqicui@uw.edu jiangyan@uw.edu zhangbao@uw.edu yyshi@eng.ucsd.edu`

## Abstract

We study the optimal control of multiple-input and multiple-output dynamical systems via the design of neural network-based controllers with stability and output tracking guarantees. While neural network-based nonlinear controllers have shown superior performance in various applications, their lack of provable guarantees has restricted their adoption in high-stake real-world applications. This paper bridges the gap between neural network-based controllers and the need for stabilization guarantees. Using equilibrium-independent passivity, a property present in a wide range of physical systems, we propose neural Proportional-Integral (PI) controllers that have provable guarantees of stability and zero steady-state output tracking error. The key structure is the strict monotonicity on proportional and integral terms, which is parameterized as gradients of strictly convex neural networks (SCNN). We construct SCNN with tunable softplus-$\beta$ activations, which yields universal approximation capability and is also useful in incorporating communication constraints. In addition, the SCNNs serve as Lyapunov functions, providing end-to-end stability and tracking guarantees. Experiments on traffic and power networks demonstrate that the proposed approach improves both transient and steady-state performances, while unstructured neural networks lead to unstable behaviors.

## 1   Introduction

Learning-based methods have the potential to solve difficult problems in control and have received significant attention from both the machine learning and control communities. A common problem in many applications is to design controllers that–with as little control effort as possible–stabilize a system at a prescribed output. A canonical example is the LQR problem and its variants, which finds optimal linear controllers for linear systems [1, 2].

For nonlinear systems, the problem is considerably harder. Neither of the two control goals, system stabilization and output tracking at the steady state, is easy to achieve. Their combination, optimizing controller costs while guaranteeing stability and output tracking at the steady state, is even more challenging. For example, vehicles in a platoon should stay in formation (stability) while cruising at the desired speed (output tracking) [3]. The optimization is then done over the set of all controllers that achieves both of these goals, e.g., reaching a platoon while consuming the least amount of fuel. Another pertinent application is the stability of electric grids with high amounts of renewables. Unlike systems with conventional generators, the power electronic interfaces of the inverters need to be actively controlled such that the grid synchronizes to the same frequency (output tracking), at the minimum operational costs. Currently, learning becomes a popular tool to parameterize nonlinear controllers as neural networks and train them to optimize performance, but provable guarantees on stability and steady-state error for these controllers are nontrivial to obtain [4–7].

Many real-world applications have multiple (possibly a continuous set of) equilibria that may be of interest. For example, a group of vehicles may need to cruise at different speeds. Moreover, the

37th Conference on Neural Information Processing Systems (NeurIPS 2023).

internal states may not be directly accessible and sometimes there are communication limit. These constraints make it difficult to enforce stability by middle steps including learning a Lyapunov function and learning a model of the system (for details, see related work in Section 1.1). Therefore, we seek to achieve "end-to-end" guarantees: the neural network-based controller should guarantee stability and steady-state error *by constructions*, for a range of possible tracking points. There are some works that showed a class of monotone controllers can provide stability guarantees for varying equilibria [8–10], but they rely on tailor-made Lyapunov functions that are found in specific applications. Especially, these results are limited to single-input and single-output (SISO) control. In practice, however, lots of complex systems need controllers that are multiple-input and multiple-output (MIMO), sometimes with high input and output dimensions.

Moreover, current learning-based approaches typically focus on optimizing transient performance (the cost and speed at which a system reaches an equilibrium), often overlooking the steady-state behavior (cost at the equilibrium state) [11–13]. In classical control terminology, these controller is proportional. Steady-state requirements are difficult to incorporate since training can only occur over a finite horizon. To enforce output tracking at the steady-state, an integral term is commonly needed to drive the system outputs to the desired value at the steady state [14–17]. This reasoning underlies the widespread use of linear Proportional-Integral (PI) controllers in practical applications [18–20]. However, linear parameterization inherently limits the controllers' degrees of freedom, potentially leading to suboptimal performance. This work addresses the following question: *Can we learn nonlinear controllers that guarantee transient stability and zero steady-state error for MIMO systems?*

**Contributions.** Clearly, it is not possible to design a controller for all nonlinear systems and the answer to the question above depends on the class of systems under consideration. This paper focuses on systems that satisfy the equilibrium independent passivity (EIP) [21, 22], which is present in many critical societal-scale systems including transportation [23], power systems [24], and communication network [25]. This abstraction allows us to design generalized controllers without considering the detailed physical system dynamics. We propose a structured Neural-PI controller that has provable guarantees of stability and zero steady-state output tracking error. The key structure we use is strictly monotone functions with vector-valued inputs and outputs. Experiments on traffic and power systems demonstrate that Neural-PI control can reduce the transient cost by at least 30% compared to the best linear controllers and maintain stability guarantees. Unstructured neural networks, on the other hand, lead to unstable behaviors. We summarize our major contributions as follows.

1) We propose a generalized PI structure for neural network-based controllers in MIMO systems, with proportional and integral terms being strictly monotone functions constructed through the gradient of strictly convex neural networks (SCNN). We construct SCNN with a tunable softplus-$\beta$ activation function and prove their universal approximation capability in Theorem 1.
2) For a multi-agent system with an underlying communication graph, we show how to restrict the controllers to respect the communication constraints through the composition of SCNNs.
3) Using EIP and the monotone functions structured by SCNNs, we design a generalized Lyapunov function that works for a range of equilibria. This provides end-to-end guarantees on asymptotic stability and zero steady-state output tracking error proved in Theorem 2. The structured neural networks can be trained for transient optimization without jeopardizing the guarantees, establishing a framework for coordinating transient optimization and steady-state requirement.

## 1.1 Related works

**Learning-based control with stability guarantees.** Recently, there has been a large interest in enforcing stability in learning-based controller [26–30]. Many works add soft penalties on the violation of stability conditions in the cost function, but it is nontrivial to certify stability for all the possible initial states in a compact set [30]. Some recent works learn a Lyapunov function and use it to certify stability through a satisfiability modulo theories (SMT) solver [26–29]. But this approach is difficult to scale to high-dimensional systems and the learned Lyapunov function only works for a single equilibrium. For control-affine systems, the work in [31] designs feedback linearizing policy with integral control to guarantee stabilizing to a range of equilibria. But it requires that all the state are accessible and many systems are not control affine. Our proposed controller guarantees stability for a set of equilibria *by construction*, and only needs access to the outputs (not the full states).

**Optimizing long-term behavior.** To regulate long-term behavior, existing works train neural networks using a cost function defined over a long time horizon [32, 33]. However, it is difficult

to quantify how long the trajectory is enough to reach the steady state, thus enforcing steady-state tracking performance by adding a long-term cost is challenging. Our proposed controller enforces steady-state tracking *by construction*, instead of relying on training.

**Algorithm to tune MIMO PI controller.** Classical Proportional Integral (PI) control structures are widely used in industrial applications, while tuning PI controller parameters in MIMO systems is tedious and relies on heuristic rules [34, 35]. Learning-based methods become popular to tune PI parameters, although still restricted to linear control gains [34, 35]. Our contribution is the more generalized PI control and the MIMO neural network for parameterization.

**Monotone neural network.** Even though scalar-valued monotone functions have been studied in the learning community [36, 37], their generalization to the vector-valued case has not. Numerous papers studied vector-valued functions that are monotone in every input, that is, $q(x) \leq q(x')$ if $x \leq x'$ [38–41]. This is only a small subclass of vector-valued monotone functions that defined as functions satisfying $(q(x) - q(x'))^\top (x - x') \geq 0$. In this paper, we construct the general class of monotone functions (often called monotone operators) [42] that satisfy the inner product inequality.

## 2 Background and Preliminaries

We consider a dynamic system described by:

$$\dot{x} = f(x, u), \quad y = h(x), \tag{1}$$

with state $x(t) \in \mathbb{R}^n$, control action $u(t) \in \mathbb{R}^m$, and output $y(t) \in \mathbb{R}^v$. We also sometimes omit the time index $t$ for simplicity[1]. In many practical applications, not all of the internal states are directly accessible. Hence, we consider control actions that follow output-feedback control laws of the form $u = g(y)$, which is a static function of the output $y$ and not the state $x$.

A state $x^*$ such that $f(x^*, u^*) = \mathbb{0}_n, y^* = h(x^*), u^* = g(y^*)$ is called an equilibrium since the system stops changing at this state. Throughout this paper, the superscript $*$ indicates the equilibrium values of variables.

**Definition 1** (Local asymptotic stability [16], Definition 4.1). *The system* (1) *is asymptotically stable around an equilibrium* $x^*$ *if,* $\forall \epsilon > 0, \exists \delta > 0$ *such that* $\|x(0) - x^*\| < \delta$ *ensures* $\|x(t) - x^*\| < \epsilon$, $\forall t \geq 0$, *and* $\exists \delta' > 0$ *such that* $\|x(0) - x^*\| < \delta'$ *ensures* $\lim_{t\to\infty} \|x(t) - x^*\| = 0$.

One of the main tools to prove asymptotic stability is the Lyapunov's direct method [27, 16]:

**Lemma 1** (Lyapunov functions and asymptotic stability [16], Theorem 4.1). *Consider the system* (1) *with an equilibrium* $x^* \in \mathcal{X} \subset \mathbb{R}^n$. *Suppose there exists a continuously differentiable function* $V : \mathcal{X} \mapsto \mathbb{R}$ *that satisfies the following conditions: 1)* $V(x^*) = 0, V(x) > 0, \forall x \in \mathcal{X} \backslash x^*$; *2)* $\dot{V}(x) = (\nabla_x V(x))^\top \dot{x} \leq 0, \forall x \in \mathcal{X}$ *with the equality holds when* $x = x^*$. *Then the system is asymptotically stable around* $x^*$.

The key challenge to using Lyapunov theory is in constructing such a function and verifying the satisfaction of the conditions in Lemma 1. An important part of this paper is to show how to systematically use neural networks to construct Lyapunov functions for a class of nonlinear systems.

Besides stability, we are often interested in achieving a specific equilibrium such that the output converges to a prescribed setpoint.

**Definition 2** (Output tracking to $\bar{y}$). *The dynamical system* (1) *is said to track a setpoint* $\bar{y}$ *if* $\lim_{t\to\infty} y(t) = \bar{y}$.

For output tracking, the dimension of the input $u$ should be at least the dimension of the output $y$, otherwise there is not enough degrees of freedom in the input to track all the outputs. If the dimensions of the input are strictly larger than the output, it is always possible to define "dummy" outputs (e.g., by appending zeros) to match the input dimension. Therefore, it is common to assume the same input and output dimensions [22]. In the remainder of this paper, we make the following assumption.

**Assumption 1** (Identical input and output dimension). *The input* $u$ *and the output* $y$ *have the same dimension, namely,* $u(t)$ *and* $y(t)$ *are both vectors in* $\mathbb{R}^m$.

---

[1]Throughout this paper, vectors are denoted in lower case bold and matrices are denoted in upper case bold, unless otherwise specified. Vectors of all ones and zeros are denoted as $\mathbb{1}_n, \mathbb{0}_n \in \mathbb{R}^n$, respectively.

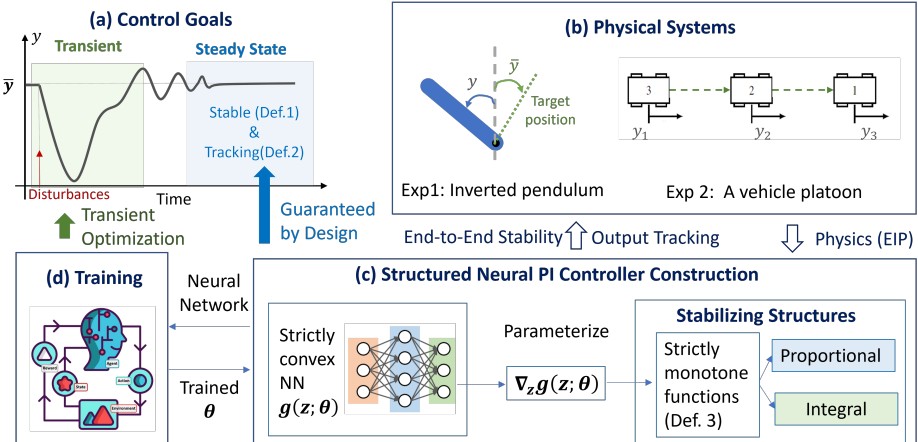

Figure 1: (a) The controller aims to improve transient performances after disturbances while guaranteeing stabilization to the desired steady-state output $\bar{y}$. (b) Two examples of physical systems with output tracking tasks. (c) We provide end-to-end stability and output tracking guarantees by enforcing stabilizing PI structure in the design of neural networks. The key structure is strictly monotone functions, which are parameterized by the gradient of SCNNs. (d) The transient performance is optimized by training the neural networks.

We will show that strictly monotone functions play an important role in guaranteeing stability and output tracking in a range of systems. Using neural networks for constructing monotone functions from $\mathbb{R}$ to $\mathbb{R}$ has been well-studied in [36, 37]. However, since we consider MIMO systems in this paper, we use a generalized notion of monotonicity for vector-valued functions as defined in [42].

**Definition 3** (Strictly monotone function [42]). *A continuous function* $\boldsymbol{q} : \mathbb{R}^m \mapsto \mathbb{R}^m$ *is strictly monotone on* $\mathcal{D} \subset \mathbb{R}^m$ *if* $(\boldsymbol{q}(\boldsymbol{\eta}) - \boldsymbol{q}(\boldsymbol{\xi}))^\top (\boldsymbol{\eta} - \boldsymbol{\xi}) \geq 0$, $\forall \boldsymbol{\eta}, \boldsymbol{\xi} \in \mathcal{D}$, *with the equality holds if and only if* $\boldsymbol{\eta} = \boldsymbol{\xi}$.

In this paper, we will show a way to construct (strictly) monotone functions using the gradient of (strictly) convex neural networks.

## 3 Problem Statement

### 3.1 Transient and steady-state requirements

We aim to optimize the controller in $\boldsymbol{u}$ to improve the transient response after disturbances while guaranteeing steady-state performance, as shown in Figure 1(a). By steady-state performance, we mean the system should settle at the desired setpoint $\bar{\boldsymbol{y}}$. By transient performance, we want the system to quickly reaching the steady state with small control efforts.

The inverted pendulum in Figure 1(b) serves as a good motivating example. The input $u$ is the force on the pendulum. The objectives include 1) the angle $y$ should reach a setpoint $\bar{y}$ (e.g., $5°$) and stays there; 2) minimizing the control cost and deviation of $y$ from $\bar{y}$ before reaching the steady state. These objectives are common in many real-world applications, such as vehicle platooning (Fig. 1(b)) and power system control in our experiments.

**Transient performance optimization.** During the transient period, our goal is to quickly stabilize the system to the desired setpoint $\bar{\boldsymbol{y}}$, while trading off with using large control efforts in $\boldsymbol{u}$. Let $J(\cdot)$ be the cost function of the system. [2] The transient optimization problem up to time $T$ is,

$$\min_{\boldsymbol{u}} \sum_{t=0}^{T} J(\boldsymbol{y}(t) - \bar{\boldsymbol{y}}, \boldsymbol{u}(t)), \tag{2a}$$

$$\text{s.t. dynamics in (1)} : \dot{\boldsymbol{x}} = \boldsymbol{f}(\boldsymbol{x}, \boldsymbol{u}), \quad \boldsymbol{y} = \boldsymbol{h}(\boldsymbol{x}), \tag{2b}$$

$$\text{stability and output tracking in Definition 1-2} : \lim_{t \to \infty} \boldsymbol{y}(t) = \bar{\boldsymbol{y}} \tag{2c}$$

Note that in (2a) we sum up to time $T$, which should be long enough to cover the transient period. Since we require tracking in (2c), the exact value of $T$ is not critical in the optimization problem. In

---

[2]Including other forms of differentiable cost functions do not change the analysis.

practice, the system dynamics (1) (and sometimes even the cost function) can be highly nonlinear and nonconvex. Therefore, the current state-of-the-art is to learn $\boldsymbol{u}$ by parameterizing it as a neural network and training it by minimizing the cost in (2a) [43, 44]. But the key challenge with applying these neural network-based controllers is guaranteeing stability and output tracking [45, 26, 30]. Even if the learned policy may appear "stable" during training, it is not necessarily stable during testing. This can be observed in both the vehicle and power system experiments in Section 6.

**Stability for a range of tracking points.** We emphasize that the setpoint $\bar{\boldsymbol{y}}$ may vary in practice (e.g., the setpoint velocity of vehicles may change), and thus the controller is required to guarantee stability and output tracking to a range of equilibria corresponding to the setpoints. This is difficult to achieve through existing works that enforce stability by learning a Lyapunov function [26, 27, 30], since the learned Lyapunov function is for a single equilibrium (normally setup as $\boldsymbol{x}^* = \mathbb{0}_n$). These methods also require that all the states $\boldsymbol{x}$ are observed, which may not be achieved in practice.

**Communication requirement.** A typical constraint in a large system is limits on communications. For example, vehicles in a group may only be able to measure the output of their neighbors (line-of-sight) or nodes in a power system may only have real-time communication from a subset of other nodes. Therefore, the control action $\boldsymbol{u}_i$ may be constrained to be a function of a subset of the outputs $\boldsymbol{y}$. We show later in Subsection 4.3 about how these communication constraints can be naturally accommodated in our controller design.

### 3.2 Bridging controller design and stability through passivity analysis

As shown in Figure 1(a), optimizing transient and steady-state performances are two problems in two different time-scales. Therefore, coordinating them is a central challenge.

**End-to-end performance guarantees.** We propose to overcome this challenge through a structured controller design that provides end-to-end stability and tracking guarantees, as shown in Figure 1(c-d). "End-to-end" means that the guarantees are provided by construction, and do not depend on how the controller is trained. Thus, the neural networks can be trained to optimize the transient performance without jeopardizing the stability and steady-state guarantees. In particular, the construction works for a range of equilibria and can conveniently incorporate communication characteristics.

Instead of working on specific systems individually, we seek to find a family of physical systems that allows us to derive a generalized controller design. It turns out that the notion of equilibrium independent passivity (EIP) [21, 22] provides a concise and useful abstraction of physical systems for stability analysis. This paper thus focuses on systems satisfying the following assumptions:

**Assumption 2** (Equilibrium-Independent Passivity [21] )**.** *The system* (1) *is strictly equilibrium-independent passive (EIP), which satisfies: (i) there exists a nonempty set $\mathcal{U}^* \subset \mathbb{R}^m$ such that for every equilibrium $\boldsymbol{u}^* \in \mathcal{U}^*$, there is a unique $\boldsymbol{x}^* \in \mathbb{R}^n$ such that $\boldsymbol{f}(\boldsymbol{x}^*, \boldsymbol{u}^*) = \mathbb{0}_n$, and (ii) there exists a positive definite storage function $S(\boldsymbol{x}, \boldsymbol{x}^*)$ and a positive scalar $\rho$ such that*

$$S(\boldsymbol{x}^*, \boldsymbol{x}^*) = 0 \quad and \quad \dot{S}(\boldsymbol{x}, \boldsymbol{x}^*) \leq -\rho \|\boldsymbol{y} - \boldsymbol{y}^*\|^2 + (\boldsymbol{y} - \boldsymbol{y}^*)^\top (\boldsymbol{u} - \boldsymbol{u}^*). \tag{3}$$

The EIP property in Assumption 2 has been found in a large class of physical systems, including transportation [23], power systems [24, 46], robotics [47], communication [25], and others. We will conduct experiments on vehicle platoons and power systems to show how EIP can be validated.

The condition (3) of EIP systems provides us a generalized approach to construct Lyapunov functions without knowing the specifies of the dynamics $\boldsymbol{f}(\cdot)$. In turn, we are able to find the right structure for the controllers. In Section 4, we show what these structures are for PI controllers and how to enforce such structures in the design of neural networks. Then Section 5 formally demonstrates the theoretical guarantees and the training procedure for transient optimization.

## 4 Structured Neural-PI Control

In this section, we construct controllers with a proportional and an integral term, which are both vector-valued strictly monotone functions parameterized by the gradient of strictly convex functions. Then, we present a neural network architecture that is strictly convex by construction and can conveniently incorporate communication constraints.

## 4.1 Generalized PI control

To realize output tracking, we introduce an integral variable $\dot{s} = \bar{y} - y$. Intuitively, $s$ is the accumulated tracking error and will remain unchanged at the steady-state when $y = \bar{y}$. On this basis, we consider a generalized PI controller $u = p(\bar{y} - y) + r(s)$. The first component is the proportional term, where $p(\bar{y} - y)$ is a function of the tracking error $\bar{y} - y$ between the current output $y$ and desired output $\bar{y}$. The second component is the integral term $r(s)$ as a function of the integral of historical tracking errors. Intuitively, the proportional term drives $y$ close to $\bar{y}$, and the integral term ensures the tracking error equals zero at the steady state.

**Generalized PI Controller.** *Compactly, the control law is*

$$u = \underbrace{p(\bar{y} - y)}_{\textit{Proportional control}} + \underbrace{r(s)}_{\textit{Integral control}} \tag{4a}$$

$$\dot{s} = \bar{y} - y. \tag{4b}$$

**Remark 1.** *The above controller can be envisioned as a nonlinear generalization of widely adopted linear Proportional-Integral (PI) controllers, where $p(\bar{y} - y) := K_P(\bar{y} - y)$, $r(s) := K_I(s)$ with $K_P$ and $K_I$ being the proportional and integral coefficients (scalar for SISO control). Linear PI control laws are almost always used in existing works [14–16], but the performance of linear PI controllers can be poor for large-scale nonlinear systems[6, 48].*

To achieve provable stability guarantees with output tracking, we need to construct a Lyapunov function that works for the closed-loop system formed by (1) and (4). Therefore, we further design structures in the proportional function $p(\cdot)$ and the integral function $r(\cdot)$ that can be utilized in Lyapunov stability analysis. The structures are strictly monotone functions, which generalizes conventional linear functions. In the next subsection, we will show how to parameterize strictly monotone functions. In section 5.1, we will show how the PI controllers structured with monotone functions provide end-to-end stability and output tracking guarantees.

## 4.2 Strictly monotone function through gradients of strictly convex neural networks

It is not trivial to parameterize strictly monotone functions since this is an infinite-dimensional function space. Scalar-valued monotone functions mapping from $\mathbb{R}$ to $\mathbb{R}$ [36, 37] has been proposed, but it is difficult to extend these designs to MIMO systems.

In this paper, we propose a new parameterization of strictly monotone functions by leveraging the fact in Proposition 1 that the gradient of a strictly convex function is strictly monotone.

**Proposition 1** (Gradients of strictly convex functions). *Let $g : \mathbb{R}^m \mapsto \mathbb{R}$ be a strictly convex function, then $(\nabla_\eta g(\eta) - \nabla_\xi g(\xi))^\top (\eta - \xi) \geq 0 \ \forall \eta, \xi \in \mathbb{R}^m$, with equality holds if and only if $\eta = \xi$. Namely, $\nabla g : \mathbb{R}^m \mapsto \mathbb{R}^m$ is a strictly monotone increasing function.*

Hence, the strictly monotone property of $p(\cdot)$ and $r(\cdot)$ can be guaranteed by design if they are the gradient of a strictly convex function, as shown by Figure 2. The next proposition shows how to construct a strictly convex neural network (SCNN).

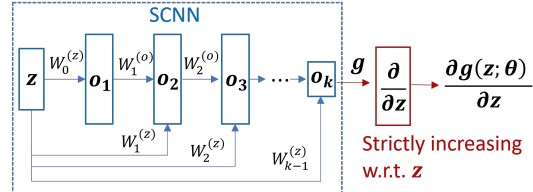

Figure 2: Strictly monotone function constructed by strictly convex neural networks (SCNN).

**Proposition 2** (Strictly convex neural networks). *Consider a function $g(z; \theta) : \mathbb{R}^m \mapsto \mathbb{R}$ parameterized by $k$-layer, fully connected neural network with the input $z \in \mathbb{R}^m$. The output $o_l$ of layer $l = 0, \ldots, k - 1$ and the function $g(z; \theta)$ is represented as*

$$o_{l+1} = \sigma_l \left( W_l^{(o)} o_l + W_l^{(z)} z + b_l \right), \quad g(z; \theta) = o_k \tag{5}$$

*where $o_0, W_0^{(o)} \equiv 0$, $\theta = \left\{ W_{0:k-1}^{(z)}, W_{1:k-1}^{(o)}, b_{0:k-1} \right\}$ are trainable weights and biases, and $\sigma_i$ are non-linear activation functions. The function $g(z; \theta)$ is strictly convex in $z$ provided that all $W_{1:k-1}^{(o)}$ are positive, $W_0^{(z)}$ is nonzero, and all functions $\sigma_l$ are strictly convex and increasing.*

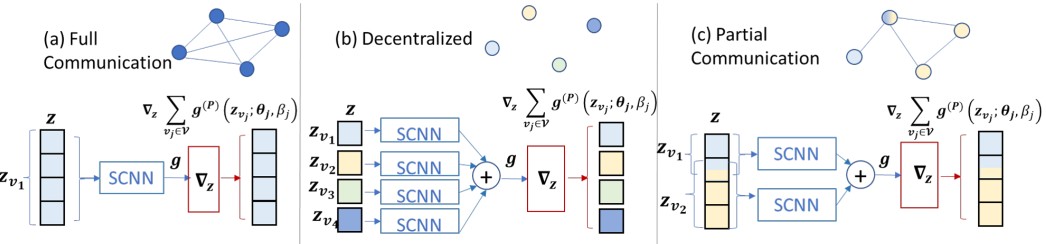

Figure 3: Communication embedded controller. Take $m = 4$ where $\boldsymbol{z} = (z_1, z_2, z_3, z_4)$ and P network $\boldsymbol{p} = \nabla_{\boldsymbol{z}} g^{(P)}(\boldsymbol{z}; \boldsymbol{\theta}^{(P)}, \beta^{(P)})$ as an example. (a) Global communication, where $\mathcal{V} = \{(1, 2, 3, 4)\}$ and $\boldsymbol{p}$ can be a function of the full $\boldsymbol{z}$. (b) Decentralized, $\mathcal{V} = \{(1), \cdots, (4)\}$ and $p_i$ is a function of $z_i, \forall i$. (c) Partial communication, $\mathcal{V} = \{(1, 2), (2, 3, 4)\}$, where there exists communication within $(1, 2)$ and $(2, 3, 4)$.

The construction of SCNN follows the general structure of input-convex neural networks [49, 50], where the conditions on activation functions and weights are modified to achieve *strictly* convexity. The proof follows from the fact that positive sums of strictly convex functions are also strictly convex and that the composition of a strictly convex and convex increasing function is also strictly convex.

The next theorem demonstrates the universal approximation property of SCNN in Proposition 2.

**Theorem 1** (Universal approximation ). *Let $\mathcal{Z}$ be a compact domain in $\mathbb{R}^m$ and $q(\boldsymbol{z}) : \mathcal{Z} \mapsto \mathbb{R}$ be a Lipschitz continuous and strictly convex function. For any $\epsilon > 0$, there exists a function $g(\boldsymbol{z}; \boldsymbol{\theta}) : \mathcal{Z} \mapsto \mathbb{R}$ constructed by (5) where the activation function is the softplus-$\beta$ function*

$$\sigma_l^{Softplus\beta}(x) := \frac{1}{\beta} \log(1 + e^{\beta x}), \tag{6}$$

*such that $|q(\boldsymbol{z}) - g(\boldsymbol{z}; \boldsymbol{\theta})| < \epsilon$ for all $\boldsymbol{z} \in \mathcal{Z}$.*

The proof is given in Appendix A.1. We sketch the proof as follows. We leverage the results in [49, 50] that the structure (5) with ReLU activation is a universal approximation for any convex function. However, ReLU activations are not strictly convex, and thus we design the Softplus-$\beta$ activation. By showing that the structure (5) with Softplus-$\beta$ activation can approximate neural networks with the ReLU activations arbitrarily closely when $\beta$ is sufficiently large, we then prove that the structure (5) with Softplus-$\beta$ can universally approximate any strictly convex function.

Notably, $\beta$ in the activation function is a tunable parameter. Hence, we write down the SCNN (5) with activation function being Softplus-$\beta$ function in (6) as $g(\boldsymbol{z}; \boldsymbol{\theta}, \beta)$, where $\beta$ is an extra trainable parameter.

Thus, we parameterize the structured Neural-PI control law $\boldsymbol{u} = \boldsymbol{p}(\bar{\boldsymbol{y}} - \boldsymbol{y}) + \boldsymbol{r}(\boldsymbol{s})$ in (4) as

$$\begin{aligned} \boldsymbol{p}(\bar{\boldsymbol{y}} - \boldsymbol{y}) &= \nabla_{\boldsymbol{z}} g^{(P)}(\boldsymbol{z}; \boldsymbol{\theta}^{(P)}, \beta^{(P)})|_{\boldsymbol{z} = \bar{\boldsymbol{y}} - \boldsymbol{y}}, \\ \boldsymbol{r}(\boldsymbol{s}) &= \nabla_{\boldsymbol{z}} g^{(I)}(\boldsymbol{z}; \boldsymbol{\theta}^{(I)}, \beta^{(I)})|_{\boldsymbol{z} = \boldsymbol{s}}. \end{aligned} \tag{7}$$

This way, $\boldsymbol{p}(\cdot)$ and $\boldsymbol{r}(\cdot)$ by construction are strictly monotone. In particular, the convex functions $g^{(P)}(\boldsymbol{z}; \boldsymbol{\theta}^{(P)}, \beta^{(P)})$, $g^{(I)}(\boldsymbol{z}; \boldsymbol{\theta}^{(I)}, \beta^{(I)})$ play a vital role in constructing a generalized Lyapunov function and showing the satisfaction of the Lyapunov condition in Subsection 5.1. This subsequently provides end-to-end stability guarantees that do not dependent on the training algorithm.

### 4.3 Embedding Communication Constraints

For some large-scale physical systems, not all of the control inputs have access to all of the output measurements at real-time. Therefore, some $u_i$'s cannot be a function of the full vector $\boldsymbol{y}$ and $\boldsymbol{s}$ because of this lack of global communication, as elaborated in Subsection 3.1.

Constructing the controllers as the gradient of convex functions provides us with a convenient way to embed these communication constraints. Let $\mathcal{V}$ be the set of indexes with communications. Take $m = 4$ where $\boldsymbol{z} = (z_1, z_2, z_3, z_4)$ and the proportional term $\boldsymbol{p} = \nabla_{\boldsymbol{z}} g^{(P)}(\boldsymbol{z}; \boldsymbol{\theta}^{(P)}, \boldsymbol{\beta}^{(P)})$ as an example. Figure 3(a) shows the case with global communication, where $\mathcal{V} = \{(1, 2, 3, 4)\}$ and $\boldsymbol{p}$ can be a function of the full $\boldsymbol{z}$. Figure 3(b) shows the case without communication where

$\mathcal{V} = \{(1), \cdots, (4)\}$ and $p_i$ can only be a function of $z_i$ for all $i$. Figure 3(c) shows the case $\mathcal{V} = \{(1,2), (2,3,4)\}$, where there exists communication within the indexes $(1,2)$ and within $(2,3,4)$. By defining SCNN $g(\boldsymbol{z}_{\boldsymbol{v}_j}; \boldsymbol{\theta}_j^{(P)}, \beta_j^{(P)})$ for each group in $\boldsymbol{v}_j \in \mathcal{V}$, the gradient $\nabla_{\boldsymbol{z}} g(\boldsymbol{z}_{\boldsymbol{v}_j}; \boldsymbol{\theta}_j^{(P)}, \beta_j^{(P)})$ will only be a function of $\boldsymbol{z}_{\boldsymbol{v}_j}$ and thus satisfying the commutation constraints. Therefore, we construct the functions in (7) as

$$\nabla_{\boldsymbol{z}} g^{(P)}(\boldsymbol{z}; \boldsymbol{\theta}^{(P)}, \boldsymbol{\beta}^{(P)}) = \nabla_{\boldsymbol{z}} \sum_{\boldsymbol{v}_j \in \mathcal{V}} g^{(P)}(\boldsymbol{z}_{\boldsymbol{v}_j}; \boldsymbol{\theta}_j^{(P)}, \beta_j^{(P)}), \tag{8}$$

where $\boldsymbol{\theta}^{(P)} := \{\boldsymbol{\theta}_1^{(P)}, \cdots, \boldsymbol{\theta}_{|\mathcal{V}|}^{(P)}\}$, $\boldsymbol{\beta}^{(P)} := \{\beta_1^{(P)}, \cdots, \beta_{|\mathcal{V}|}^{(P)}\}$, and $\boldsymbol{\theta}_j^{(P)}, \boldsymbol{\theta}_j^{(I)}$ are parameters of the SCNNs for group $\boldsymbol{v}_j$ within the communication network.

The function $g^{(I)}(\boldsymbol{z}; \boldsymbol{\theta}^{(I)}, \beta^{(I)})$ is also constructed in a similar way as $g^{(P)}(\boldsymbol{z}; \boldsymbol{\theta}^{(P)}, \beta^{(P)})$ in (8) to incorporate the communication constraints in $\mathcal{V}$. Strict convexity still holds since a sum of strictly convex functions is also strictly convex.

# 5 Training with End-to-End Guarantees

## 5.1 End-to-end stability and output-tracking guarantees

The convex function $g^{(I)}(\boldsymbol{s}; \boldsymbol{\theta}^{(I)}, \boldsymbol{\beta}^{(I)})$ constructed from SCNNs can be utilized to construct a Lyapunov function for proving stability using the Bregman distance defined in the following Lemma [51]:

**Lemma 2** (Bregman distance). *The Bregman distance associated with the convex function $g^{(I)}(\boldsymbol{s}; \boldsymbol{\theta}^{(I)}, \boldsymbol{\beta}^{(I)})$ is defined by*

$$B(\boldsymbol{s}, \boldsymbol{s}^*; \boldsymbol{\theta}^{(I)}, \boldsymbol{\beta}^{(I)}) = g^{(I)}(\boldsymbol{s}; \boldsymbol{\theta}^{(I)}, \boldsymbol{\beta}^{(I)}) - g^{(I)}(\boldsymbol{s}^*; \boldsymbol{\theta}^{(I)}, \boldsymbol{\beta}^{(I)}) - \nabla_{\boldsymbol{s}} g^{(I)}(\boldsymbol{s}^*; \boldsymbol{\theta}^{(I)}, \boldsymbol{\beta}^{(I)})^\top (\boldsymbol{s} - \boldsymbol{s}^*),$$

*which is positive definite with equality holds if and only if $\boldsymbol{s} = \boldsymbol{s}^*$.*

The Bregman distance allows us to construct Lyapunov functions without specifying the equilibrium $\boldsymbol{s}^*$. Combining the storage function in Assumption 2 and $B(\boldsymbol{s}, \boldsymbol{s}^*; \boldsymbol{\theta}^{(I)}, \beta^{(I)})$ in Lemma 2, next theorem shows that the Neural-PI controller stabilizes the system to the desired output $\bar{\boldsymbol{y}}$.

**Theorem 2.** *Suppose Assumption 2 holds and the input $\boldsymbol{u}$ follows the structured PI control law $\boldsymbol{u} = \boldsymbol{p}(\bar{\boldsymbol{y}} - \boldsymbol{y}) + \boldsymbol{r}(\boldsymbol{s})$ where $\boldsymbol{p}(\cdot)$ and $\boldsymbol{r}(\cdot)$ are constructed as the gradients of strictly convex neural networks in (7). If the system (1) has a feasible equilibrium, then the system is locally asymptotically stable around it. In particular, the steady-state outputs track the setpoint $\bar{\boldsymbol{y}}$, namely $\boldsymbol{y}^* = \bar{\boldsymbol{y}}$.*

Theorem 2 shows that Neural-PI control has provable guarantees on stability and zero steady-state output tracking error from the structures in (7). Detailed proof could be found in Appendix A.2 and we sketch the proof below. At an equilibrium, the right side of (4b) equals to zero gives $\boldsymbol{y}^* = \bar{\boldsymbol{y}}$. We show that an equilibrium is asymptotically stable by constructing a Lyapunov function $V(\boldsymbol{x}, \boldsymbol{s})|_{\boldsymbol{x}^*, \boldsymbol{s}^*} = S(\boldsymbol{x}, \boldsymbol{x}^*) + B(\boldsymbol{s}, \boldsymbol{s}^*; \boldsymbol{\theta}^{(I)}, \boldsymbol{\beta}^{(I)})$. Since $\boldsymbol{r}(\boldsymbol{s}) = \nabla_{\boldsymbol{s}} g^{(I)}(\boldsymbol{s}; \boldsymbol{\theta}^{(I)}, \beta^{(I)})$ by construction in (7), the time derivative of the Bregman distance term is $\dot{B}(\boldsymbol{s}, \boldsymbol{s}^*; \boldsymbol{\theta}^{(I)}, \boldsymbol{\beta}^{(I)}) = (\boldsymbol{r}(\boldsymbol{s}) - \boldsymbol{r}(\boldsymbol{s}^*))^\top (-(\boldsymbol{y} - \boldsymbol{y}^*))$. Combining with the fact $\dot{S}(\boldsymbol{x}, \boldsymbol{x}^*) \leq -\rho \|\boldsymbol{y} - \boldsymbol{y}^*\|^2 + (\boldsymbol{y} - \boldsymbol{y}^*)^\top (\boldsymbol{u} - \boldsymbol{u}^*)$ (from the EIP assumption), and $\boldsymbol{u} = \boldsymbol{p}(\bar{\boldsymbol{y}} - \boldsymbol{y}) + \boldsymbol{r}(\boldsymbol{s})$, we can conclude that the Lyapunov stability condition holds.

**Remark 2.** *The satisfaction of the Lyapunov condition does not depend on the specifics of $\boldsymbol{f}(\cdot)$ (as long as it is EIP), making the stability certification robust to parameter changes for systems satisfying Assumption 2. We will demonstrate this in the experiment on power system control.*

## 5.2 Training to improve transient performances

The Neural-PI controller can be trained by most model-based or model-free algorithms, since the stability and output-tracking are guaranteed by construction. Fig 4 visualizes the detailed construction and the computation graph in the dynamical system in (1). The trainable parameters $\boldsymbol{\Theta} := \{\boldsymbol{\theta}^{(P)}, \boldsymbol{\beta}^{(P)}, \boldsymbol{\theta}^{(I)}, \boldsymbol{\beta}^{(I)}\}$ are contained in $\boldsymbol{p}(\cdot)$ and $\boldsymbol{r}(\cdot)$ functions, where both are parameterized as the gradients of SCNNs. $\boldsymbol{u} = \nabla_{\bar{\boldsymbol{y}} - \boldsymbol{y}} g^{(P)}(\bar{\boldsymbol{y}} - \boldsymbol{y}; \boldsymbol{\theta}^{(P)}, \beta^{(P)}) + \nabla_{\boldsymbol{s}} g^{(I)}(\boldsymbol{s}; \boldsymbol{\theta}^{(I)}, \beta^{(I)})$ serves as control signal in the system defined in (1) that evolves through time. The loss function is defined as

$$Loss(\boldsymbol{\Theta}) = \sum_{t=0}^{T} J(\boldsymbol{y}(t) - \bar{\boldsymbol{y}}, \boldsymbol{u}(t)), \tag{9}$$

where $J(\cdot)$ is the transient cost function defined in (2a). The parameters $\Theta$ are optimized via gradient descent to minimize this loss function (9).

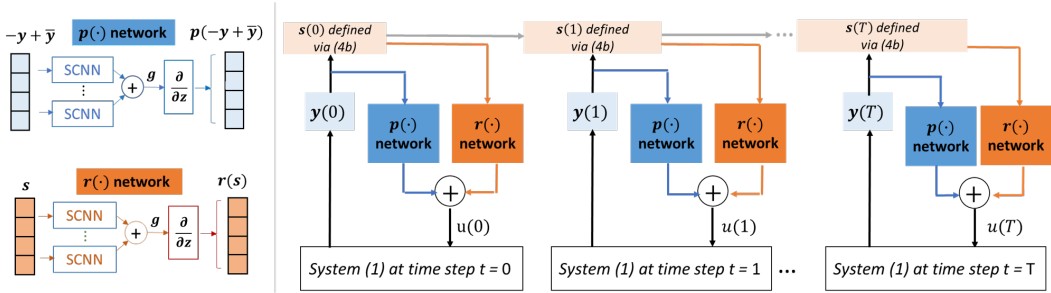

Figure 4: The computation graph for training the Neural-PI controllers.

## 6 Experiments

We end the paper with case studies demonstrating the effectiveness of the proposed Neural-PI control in two large-scale systems: vehicle platooning and power system frequency control. Detailed problem formulation, verification of assumptions, simulation setting, and results are provided in Appendix B.1 and B.2 in the supplementary material. All experiments are run with an NVIDIA Tesla T4 GPU with 16GB memory. Code is available at this link.

### 6.1 Vehicle platooning

**Experiment setup.** We conduct experiments on the vehicle platooning problem in Figure 1(b) with 20 vehicles ($m = 20$), where $\boldsymbol{u} \in \mathbb{R}^m$ is the control signal to adjust the velocities of vehicles and the output $\boldsymbol{y} \in \mathbb{R}^m$ is their actual velocities. The state is $\boldsymbol{x} = (\boldsymbol{\zeta}, \boldsymbol{y})$, where $\boldsymbol{\zeta} \in \mathbb{R}^m$ is the relative position of vehicles. The dynamic model is $\dot{\boldsymbol{\zeta}} = \boldsymbol{\Gamma} \boldsymbol{y}$, $\dot{\boldsymbol{y}} = \hat{\boldsymbol{\kappa}}(-(\boldsymbol{y} - \boldsymbol{\lambda}^0) + \hat{\boldsymbol{\rho}}(\boldsymbol{u} - \boldsymbol{E}\hat{\boldsymbol{D}}\boldsymbol{E}^\top\boldsymbol{\zeta}))$, where $\hat{\boldsymbol{\kappa}}, \hat{\boldsymbol{\rho}}, \hat{\boldsymbol{D}}, \boldsymbol{E}, \boldsymbol{\Gamma}$ are constant matrices with their physical meaning given in Appendix B.1.1. The vector $\boldsymbol{\lambda}^0 \in \mathbb{R}^m$ reflects the default velocity of vehicles. In Appendix B.1.2, We verify that this system is EIP (i.e., satisfying Assumption 2) using the storage function $S(\boldsymbol{x}, \boldsymbol{x}^*) = \frac{1}{2}(\boldsymbol{y} - \boldsymbol{y}^*)^\top\hat{\boldsymbol{\kappa}}^{-1}\hat{\boldsymbol{\rho}}^{-1}(\boldsymbol{y} - \boldsymbol{y}^*) + \frac{1}{2}\boldsymbol{\zeta}^\top\boldsymbol{E}\hat{\boldsymbol{D}}\boldsymbol{E}^\top\boldsymbol{\zeta}$. The objective is for the vehicles to reach the same setpoint velocity quickly with acceptable control effort. We train for 400 epochs, where each epoch trains with the loss (9) averaged on 300 trajectories, and each trajectory evolves 6s from random initial velocities.

**Controller performance.** We compare the performance of 1) Neural-PI: the learned structured Neural-PI controllers parametrized by (7) with three layers and 20 neurons in each hidden layer. 2) DenseNN: Dense neural networks (5) the same as Neural-PI with unconstrained weights. Both of them have no communication constrints. 3) Linear-PI: linear PI control where $\boldsymbol{p}(\bar{\boldsymbol{y}} - \boldsymbol{y}) := \boldsymbol{K}_P(\bar{\boldsymbol{y}} - \boldsymbol{y})$, $\boldsymbol{r}(\boldsymbol{s}) := \boldsymbol{K}_I(\boldsymbol{s})$ with $\boldsymbol{K}_P$ and $\boldsymbol{K}_I$ being the trainable proportional and integral coefficients. Figure 5(a) shows the transient and steady-state costs on 100 testing trajectories starting from randomly generated initial states. Neural-PI attains much lower costs even though the weights in DenseNN are not constrained. Compared with Linear-PI, Neural-PI achieves similar steady-state cost (since it retains all the stability guarantees of classical linear PI control), but reduces the transient cost by approximately 30%. Given $\bar{\boldsymbol{y}} = 5$m/s, Figure 5(b) and 5(c) show the dynamics of selected nodes under DenseNN and Neural-PI, respectively. All the outputs track under Neural-PI but they are unstable under DenseNN (even though the training cost was not prohibitively high). In particular, DenseNN appears to be stable until about 10s, but states blows up quickly after that. Therefore, enforcing stabilizing structures is essential.

### 6.2 Power systems frequency control

**Experiment Setup.** The second experiment is the power system frequency control on the IEEE 39-bus New England system [52], where $\boldsymbol{u} \in \mathbb{R}^m$ is the control signal to adjust the power injection from generators and the output $\boldsymbol{y} \in \mathbb{R}^m$ is the rotating speed (i.e., frequency) of generators. The objective is to stabilize generators at the required frequency $\bar{\boldsymbol{y}} = 60$Hz at the steady state while minimizing the transient control cost. The state is $\boldsymbol{x} = (\boldsymbol{\delta}, \boldsymbol{y})$, where $\boldsymbol{\delta} \in \mathbb{R}^m$ is the rotating angle of

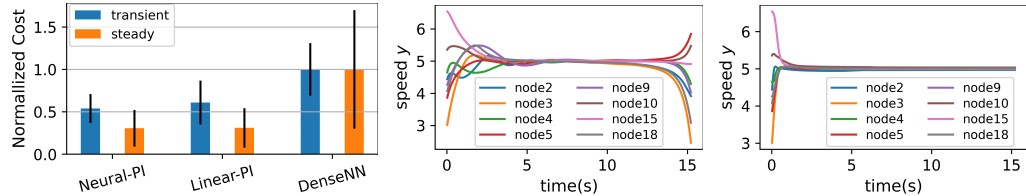

(a) Transient and steady-state cost     (b) Velocities under DenseNN     (c) Velocities under Neural-PI

Figure 5: (a) The average transient cost and steady-state cost with error bar on 100 testing trajectories starting from randomly generated initial states. (b) The dynamics of DenseNN. (c) The dynamics of Neural-PI.

generators. The dynamics of the system is $\dot{\boldsymbol{\delta}} = \boldsymbol{\Gamma} \boldsymbol{y}$, $\hat{\boldsymbol{M}} \dot{\boldsymbol{y}} = -\hat{\boldsymbol{D}}(\boldsymbol{y} - \bar{\boldsymbol{y}}) - \boldsymbol{d} + \boldsymbol{u} - \boldsymbol{E}\hat{\boldsymbol{b}}\sin(\boldsymbol{E}^\top \boldsymbol{\delta})$, where $\hat{\boldsymbol{M}}$, $\hat{\boldsymbol{D}}$ $\hat{\boldsymbol{b}}$, $\boldsymbol{E}$, $\boldsymbol{\Gamma}$ are constant matrices with their physical meaning given in Appendix B.2.1. The vector $\boldsymbol{d}$ is the net load of the system. In Appendix B.2.2, We verify that this system is EIP (i.e., satisfying Assumption 2) using the storage function $S(\boldsymbol{x}, \boldsymbol{x}^*) = \frac{1}{2}(\boldsymbol{y} - \boldsymbol{y}^*)^\top \hat{\boldsymbol{M}}(\boldsymbol{y} - \boldsymbol{y}^*) - \mathbb{1}_e^\top \hat{\boldsymbol{b}}(\cos(\boldsymbol{E}^\top \boldsymbol{\delta}) - \cos(\boldsymbol{E}^\top \boldsymbol{\delta}^*)) - (\boldsymbol{E}\hat{\boldsymbol{b}}\sin(\boldsymbol{E}^\top \boldsymbol{\delta}^*))^\top(\boldsymbol{\delta} - \boldsymbol{\delta}^*))$. Below we show the impact of communication constraints on the performance of Neural-PI control. More simulation results can be found in Appendix B.2.3 to demonstrate 1) Neural-PI control is robust to parameter changes, disturbances and noises. 2) Neural-PI significantly reduces the number of sampled trajectories to train well compared with DenseNN.

**Impact of communication constraints.** Most systems do not have fully connected real-time communication capabilities, so the controller needs to respect the communication constraints and we show the flexibility of Neural-PI control under different communication structures. We compare the performance of Neural-PI controller where 1) all the nodes can communicate 2) half of the nodes can communicate and 3) none of the nodes can communicate (thus the controller is decentralized), respectively. The transient and steady-state costs are illustrated in Figure 6(a). Neural-PI-Full achieves the lowest transient and steady-state cost. Notably, the steady-state cost significantly decreases with increased communication capability. The reason is that communication serves to better allocated control efforts such that they can maintain output tracking with smaller control costs. The frequency dynamics are shown in Figure 6(b)-(d), where under all settings Neural-PI controllers can stabilize to the setpoint (60Hz) and it converges the fastest under full communication.

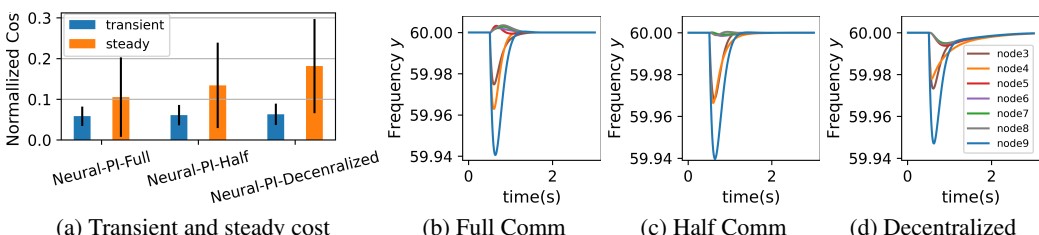

(a) Transient and steady cost    (b) Full Comm    (c) Half Comm    (d) Decentralized

Figure 6: (a) The average transient and steady-state cost with error bar on 100 testing trajectories for Neural-PI controller with different communication constraints. The dynamics of Neural-PI where (b) all nodes can communicate, (c) half of nodes can communicate, (d) none nodes can communicate.

# 7 Conclusions

This paper proposes structured Neural-PI controllers for multiple-input multiple-output dynamic systems. We parameterize the P and I terms as the gradients of strictly convex neural networks. For a wide range of dynamic systems, we show that this controller structure provides end-to-end stability and zero output tracking error guarantees. Experiments demonstrate that the Neural-PI control law retains all the stability guarantees of classical linear PI control, but achieves much lower transient cost. Unstructured neural networks, however, lead to unstable behavior and much higher costs. The theoretic guarantees of Neural-PI control also significantly reduce the amount of data required to train well. Since classical PI control is widely utilized in real-world applications, we expect that the controllers can be transferred to real-world scenarios. Potential barriers to the application in real-world scenarios include the verification of EIP when a storage function is difficult to find and provable guarantees on the robustness to noises. These are all important future directions for us.

## Acknowledgments and Disclosure of Funding

The authors are partially supported by the NSF grants ECCS-1930605, 1942326, 2200692, and 2153937.

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

# A  Proof

## A.1  Proof of Theorem 1

We leverage the results in [49, 50] that the structure (5) with ReLU activation is a universal approximation for any convex function. However, ReLU activations are not strictly convex, and thus we design the Softplus-$\beta$ activation. By showing that the structure (5) with Softplus-$\beta$ activation can approximate neural networks with the ReLU activations arbitrarily closely when $\beta$ is sufficiently large, we then prove that the structure (5) with Softplus-$\beta$ can universally approximate any strictly convex function.

To prepare for the proof of Theorem 1, we first derive the following Lemma about the difference between ReLU activations and Softplus-$\beta$ activation.

**Lemma 3.** *Consider the ReLU activation $\sigma_l^{ReLU}(x) := \max(x, 0)$. For all $x \in \mathbb{R}$ and $\Delta > 0$, we have $0 < \left( \sigma_l^{Softplus\beta}(x + \Delta) - \sigma_l^{ReLU}(x) \right) < \Delta + \frac{1}{\beta} \log(2)$.*

*Proof.* Note that

$$\left( \sigma_l^{\text{Softplus}\beta}(x + \Delta) - \sigma_l^{\text{ReLU}}(x) \right) = \left( \sigma_l^{\text{Softplus}\beta}(x + \Delta) - \sigma_l^{\text{Softplus}\beta}(x) \right) + \left( \sigma_l^{\text{Softplus}\beta}(x) - \sigma_l^{\text{ReLU}}(x) \right),$$

we prove the lemma by deriving the bound for $\left( \sigma_l^{\text{Softplus}\beta}(x + \Delta) - \sigma_l^{\text{Softplus}\beta}(x) \right)$ and $\left( \sigma_l^{\text{Softplus}\beta}(x) - \sigma_l^{\text{ReLU}}(x) \right)$ as follows

(i) Since $\frac{d\sigma_l^{\text{Softplus}\beta}(x)}{dx} = e^{\beta x}/(1 + e^{\beta x}) \in (0, 1)$, then $0 < \left( \sigma_l^{\text{Softplus}\beta}(x + \Delta) - \sigma_l^{\text{Softplus}\beta}(x) \right) < \Delta$ for $\Delta > 0$.

(ii) Explicitly represent $\sigma_l^{\text{Softplus}\beta}(x) - \sigma_l^{\text{ReLU}}(x)$ yields

$$\sigma_l^{\text{Softplus}\beta}(x) - \sigma_l^{\text{ReLU}}(x) = \begin{cases} \frac{1}{\beta} \log(1 + e^{\beta x}) - x, & x \geq 0 \\ \frac{1}{\beta} \log(1 + e^{\beta x}), & x < 0 \end{cases}$$

Thus,

$$\frac{d\left( \sigma_l^{\text{Softplus}\beta}(x) - \sigma_l^{\text{ReLU}}(x) \right)}{dx} := \begin{cases} -1/(1 + e^{\beta x}), & x \geq 0 \\ e^{\beta x}/(1 + e^{\beta x}), & x < 0 \end{cases}$$

and therefore $\left( \sigma_l^{\text{Softplus}\beta}(x) - \sigma_l^{\text{ReLU}}(x) \right) \leq \left( \sigma_l^{\text{Softplus}\beta}(0) - \sigma_l^{\text{ReLU}}(0) \right) = \frac{1}{\beta} \log(2)$. Note that $\frac{1}{\beta} \log(1 + e^{\beta x}) > \frac{1}{\beta} \log(e^{\beta x}) = x$, then $0 < \left( \sigma_l^{\text{Softplus}\beta}(x) - \sigma_l^{\text{ReLU}}(x) \right) \leq \frac{1}{\beta} \log(2)$.

Combining (i) and (ii), $0 < \left( \sigma_l^{\text{Softplus}\beta}(x + \Delta) - \sigma_l^{\text{ReLU}}(x) \right) < \Delta + \frac{1}{\beta} \log(2)$.

$\square$

The proof of Theorem 1 is given below.

*Proof.* Previous works [49, 50] have shown that the structure (5) with ReLU activation is a universal approximation for any convex function. Hence, for any $q(z) : \mathcal{Z} \mapsto \mathbb{R}$ , there exists $g(z; \boldsymbol{\theta})^{\text{ReLU}}$ constructed by (5) where the activation function is ReLU and satisfying $|g(z; \boldsymbol{\theta})^{\text{ReLU}} - q(z)| < \frac{1}{2}\epsilon$. Note that

$$|g(z; \boldsymbol{\theta})^{\text{Softplus}\beta} - q(z)| \leq |g(z; \boldsymbol{\theta})^{\text{Softplus}\beta} - g(z; \boldsymbol{\theta})^{\text{ReLU}}| + |g(z; \boldsymbol{\theta})^{\text{ReLU}} - q(z)|,$$

then it suffices to prove $|g(z; \boldsymbol{\theta})^{\text{Softplus}\beta} - q(z)| < \epsilon$ by showing the existence of $g(z; \boldsymbol{\theta})^{\text{Softplus}\beta}$ such that $|g(z; \boldsymbol{\theta})^{\text{Softplus}\beta} - g(z; \boldsymbol{\theta})^{\text{ReLU}}| < \frac{1}{2}\epsilon$ for all $z \in \mathcal{Z}$.

Next, we compute the difference of the structure (5) with softplus-$\beta$ and ReLU activations by tracing through the first layer to the last layer, under the same weights $\boldsymbol{\theta} = \left\{ W_{0:k-1}^{(z)}, W_{1:k-1}^{(o)}, b_{0:k-1} \right\}$.

The difference between the output of the first layer in $g(\boldsymbol{z};\boldsymbol{\theta})^{\text{Softplus}\beta}$ and $g(\boldsymbol{z};\boldsymbol{\theta})^{\text{ReLU}}$ is

$$\boldsymbol{o}_1^{\text{Softplus}\beta} - \boldsymbol{o}_1^{\text{ReLU}} = \sigma_1^{\text{Softplus}\beta}\left(W_0^{(z)}\boldsymbol{z} + b_0\right) - \sigma_1^{\text{ReLU}}\left(W_0^{(z)}\boldsymbol{z} + b_0\right), \tag{10}$$

which yields $\mathbb{0} < \boldsymbol{o}_1^{\text{Softplus}\beta} - \boldsymbol{o}_1^{\text{ReLU}} \leq \frac{1}{\beta}\log(2)\mathbb{1}$ by Lemma 3.

The difference of the second layer is

$$\begin{aligned}
\boldsymbol{o}_2^{\text{Softplus}\beta} - \boldsymbol{o}_2^{\text{ReLU}} &= \sigma_2^{\text{Softplus}\beta}\left(W_1^{(o)}\boldsymbol{o}_1^{\text{Softplus}\beta} + W_2^{(z)}\boldsymbol{z} + b_2\right) - \sigma_2^{\text{ReLU}}\left(W_1^{(o)}\boldsymbol{o}_1^{\text{ReLU}} + W_2^{(z)}\boldsymbol{z} + b_2\right) \\
&= \sigma_2^{\text{Softplus}\beta}\left(W_1^{(o)}\left(\boldsymbol{o}_1^{\text{Softplus}\beta} - \boldsymbol{o}_1^{\text{ReLU}}\right) + W_1^{(o)}\boldsymbol{o}_1^{\text{ReLU}} + W_2^{(z)}\boldsymbol{z} + b_2\right) \\
&\quad - \sigma_2^{\text{ReLU}}\left(W_1^{(o)}\boldsymbol{o}_1^{\text{ReLU}} + W_2^{(z)}\boldsymbol{z} + b_2\right).
\end{aligned} \tag{11}$$

Since all the element of $W_1^{(o)}$ is positive, we have $\mathbb{0} < W_1^{(o)}\left(\boldsymbol{o}_1^{\text{Softplus}\beta} - \boldsymbol{o}_1^{\text{ReLU}}\right) \leq \frac{1}{\beta}\log(2)W_1^{(o)}\mathbb{1}$.

Applying Lemma 3 element-wise yields $\mathbb{0} \leq \boldsymbol{o}_2^{\text{Softplus}\beta} - \boldsymbol{o}_2^{\text{ReLU}} \leq \frac{1}{\beta}\log(2)W_1^{(o)}\mathbb{1} + \frac{1}{\beta}\log(2)\mathbb{1}$.

Similarily $\mathbb{0} \leq \boldsymbol{o}_3^{\text{Softplus}\beta} - \boldsymbol{o}_3^{\text{ReLU}} \leq \frac{1}{\beta}\log(2)W_2^{(o)}W_1^{(o)}\mathbb{1} + \frac{1}{\beta}\log(2)W_2^{(o)}\mathbb{1} + \frac{1}{\beta}\log(2)\mathbb{1}$. By induction

$$\mathbb{0} \leq \boldsymbol{o}_{l+1}^{\text{Softplus}\beta} - \boldsymbol{o}_{l+1}^{\text{ReLU}} \leq \frac{1}{\beta}\log(2)\left(\mathbb{1} + \left(\sum_{i=1}^{l}\prod_{j=i}^{l} W_j^{(o)}\mathbb{1}\right)\right) \tag{12}$$

Hence,

$$0 \leq g(\boldsymbol{z};\boldsymbol{\theta})^{\text{Softplus}\beta} - g(\boldsymbol{z};\boldsymbol{\theta})^{\text{ReLU}} \leq \frac{1}{\beta}\log(2)\left(1 + \left(\sum_{i=1}^{k-1}\prod_{j=i}^{k-1} W_j^{(o)}\mathbb{1}\right)\right), \tag{13}$$

where $\prod_{j=i}^{k-1} W_j^{(o)} := W_{k-1}^{(o)}W_{k-2}^{(o)}\cdots W_i^{(o)}$.

Let $\beta > \frac{2}{\epsilon}\log(2)\left(1 + \left(\sum_{i=1}^{k-1}\prod_{j=i}^{k-1} W_j^{(o)}\right)\mathbb{1}\right)$, then $0 \leq g(\boldsymbol{z};\boldsymbol{\theta})^{\text{Softplus}\beta} - g(\boldsymbol{z};\boldsymbol{\theta})^{\text{ReLU}} \leq \frac{1}{2}\epsilon$. We complete the proof using

$$|g(\boldsymbol{z};\boldsymbol{\theta})^{\text{Softplus}\beta} - q(\boldsymbol{z})| \leq |g(\boldsymbol{z};\boldsymbol{\theta})^{\text{Softplus}\beta} - g(\boldsymbol{z};\boldsymbol{\theta})^{\text{ReLU}}| + |g(\boldsymbol{z};\boldsymbol{\theta})^{\text{ReLU}} - q(\boldsymbol{z})| < \frac{1}{2}\epsilon + \frac{1}{2}\epsilon = \epsilon.$$

$\square$

### A.2  Proof of Theorem 2

*Proof.* At an equilibrium, the right side of (4b) equals to zero gives $\boldsymbol{y}^* = \bar{\boldsymbol{y}}$ and the corresponding set of equlibrium $\mathcal{E} = \{\boldsymbol{x}^*, \boldsymbol{s}^* | \boldsymbol{f}(\boldsymbol{x}^*, \boldsymbol{r}(\boldsymbol{s}^*)) = \boldsymbol{0}, \boldsymbol{y}^* = \bar{\boldsymbol{y}}, \boldsymbol{h}(\boldsymbol{x}^*) = \boldsymbol{y}^*\}$. We construct a Lyapunov function to prove that if there is a feasible equilibrium in $\mathcal{E}$, then the system is locally asymptotically stable around it.

We construct a Lyapunov function using the storage function in Assumption 2 and the Bregman distance in Lemma 2 as

$$V(\boldsymbol{x}, \boldsymbol{s})|_{\boldsymbol{x}^*, \boldsymbol{s}^*} = S(\boldsymbol{x}, \boldsymbol{x}^*) + B(\boldsymbol{s}, \boldsymbol{s}^*; \boldsymbol{\theta}^{(I)}, \boldsymbol{\beta}^{(I)}), \tag{14}$$

where the functions by construction satisfy $S(\boldsymbol{x}, \boldsymbol{x}^*) \geq 0$, $B(\boldsymbol{s}, \boldsymbol{s}^*; \boldsymbol{\theta}^{(I)}, \boldsymbol{\beta}^{(I)}) \geq 0$ with equality holds only when $\boldsymbol{x} = \boldsymbol{x}^*$ and $\boldsymbol{s} = \boldsymbol{s}^*$, respectively. Hence, $V(\boldsymbol{x}, \boldsymbol{s})|_{\boldsymbol{x}^*, \boldsymbol{s}^*}$ is a well-defined function that is positive definite and equals to zero at the equilibrium.

To prepare for the calculation of the time derivative of the Lyapunov function, we start by calculating the time derivative of the function $B(\boldsymbol{s}, \boldsymbol{s}^*; \boldsymbol{\theta}^{(I)}, \boldsymbol{\beta}^{(I)})$:

$$\begin{aligned}
\dot{B}(\boldsymbol{s}, \boldsymbol{s}^*; \boldsymbol{\theta}^{(I)}, \boldsymbol{\beta}^{(I)}) &= \left(\nabla_{\boldsymbol{s}} g^{(I)}(\boldsymbol{s}; \boldsymbol{\theta}^{(I)}, \boldsymbol{\beta}^{(I)}) - \nabla_{\boldsymbol{s}} g^{(I)}(\boldsymbol{s}^*; \boldsymbol{\theta}^{(I)}, \boldsymbol{\beta}^{(I)})\right)^\top \dot{\boldsymbol{s}} \\
&\overset{①}{=} (\boldsymbol{r}(\boldsymbol{s}) - \boldsymbol{r}(\boldsymbol{s}^*))^\top (-(\boldsymbol{y} - \boldsymbol{y}^*)),
\end{aligned} \tag{15}$$

where ① follows from $\nabla_s g^{(I)}(s; \theta^{(I)}, \beta^{(I)}) = r(s)$ and $\dot{s} = (-(y - \bar{y})) = (-(y - y^*))$.

The time derivative of the Lyapunov function is

$$
\begin{aligned}
\dot{V}(x, s)|_{x^*, s^*} =& \dot{S}(x, x^*) + \dot{B}(s, s^*; \theta, \beta) \\
&\overset{①}{\leq} -\rho \|y - y^*\|^2 + (y - y^*)^\top (u - u^*) + (r(s) - r(s^*))^\top (-(y - y^*)) \\
&\overset{②}{=} -\rho \|y - y^*\|^2 + (y - y^*)^\top (p(-y + \bar{y}) - p(-y^* + \bar{y})) \\
&\overset{③}{\leq} -\rho \|y - y^*\|^2
\end{aligned}
\tag{16}
$$

where ① follows from the strict EIP property and equations derived in (15). The equality ② is derived by plugging in the controller design in (4a) where $u = p(-y + \bar{y}) + r(s)$ and $u^* = p(-y^* + \bar{y}) + r(s^*)$. The inequality ③ uses strictly monotone property of $p(\cdot)$.

Therefore, $\dot{V}(x, s)|_{x^*, s^*} \leq 0$ with equality only holds at the equilibrium. By Lyapunov stability theory in Proposition 1, the system is locally asymptotically stable around the equilibrium.

$\square$

# B  Experiments

We demonstrate the effectiveness of the proposed neural-PI control in two dynamical systems: vehicle platooning and power system frequency control. All experiments are run with an NVIDIA Tesla T4 GPU with 16GB memory. For completeness, the figures highlighted in Section 6 are also shown below with more thorough discussions. Code is available at this link.

## B.1  Vehicle platooning

### B.1.1  Problem statement

The first experiment is the vehicle platoon control in [3, 23] with $m$ vehicles, where $u \in \mathbb{R}^m$ is the control signal to adjust the velocities of vehicles and the output $y \in \mathbb{R}^m$ is their actual velocities. The state is $x = (\zeta, y)$, where $\zeta \in \mathbb{R}^m$ is the relative position of vehicles with $\zeta(0) \perp Im(\mathbb{1}_m)$ (namely, the vehicles are not in the same position at the time step 0). The dynamic model is

$$
\begin{aligned}
\dot{\zeta} &= \Gamma y, \\
\dot{y} &= \hat{\kappa} \left( -(y - \lambda^0) + \hat{\rho} \left( u - E \hat{D} E^\top \zeta \right) \right),
\end{aligned}
\tag{17}
$$

where $\hat{\kappa} = \mathrm{diag}(\kappa_1, \cdots, \kappa_m), \hat{\rho} = \mathrm{diag}(\rho_1, \cdots, \rho_m) \in \mathbb{R}^{m \times m}$ are constant diagonal matrices and $\kappa_i > 0, \rho_i > 0$ for all $i = 1, \cdots, m$. The vector $\lambda^0 = (\lambda_1^0, \cdots, \lambda_m^0) \in \mathbb{R}^m$ reflects the default velocity of vehicles. The matrix $E \in \mathbb{R}^{m \times e}$ is the incidence matrix that indicates the neighbouring relations for $e$ pairs of neighbouring vehicles and satisfy $ker(E^\top) = Im(\mathbb{1}_m)$. The matrix $\Gamma := I_m - \frac{1}{m} \mathbb{1}_m \mathbb{1}_m^\top$ extracts the relative velocities of vehicles by $\Gamma y$. The diagonal matrix $\hat{D} \in \mathbb{R}^{m \times m}$ represents the sensitivity to the relative distance of neighbouring vehicles.

### B.1.2  Verification of Assumption 2

We start by verifying the uniqueness of $x^*$ for any $u^* \in \mathbb{R}^m$. At the equilibrium, the right side of (17) equals zero gives

$$
\left( -(y^* - \lambda^0) + \hat{\rho} \left( u^* - E \hat{D} E^\top \zeta^* \right) \right) = \mathbb{0}_m \text{ and } \Gamma y^* = \mathbb{0}_m.
\tag{18}
$$

For a given $u^* \in \mathbb{R}^m$, suppose there exists $x_a^* = (\zeta_a^*, y_a^*)$ and $x_b^* = (\zeta_b^*, y_b^*)$, $x_a^* \neq x_b^*$ such that (18) holds. Plugging in (18) gives

$$
\begin{aligned}
(y_a^* - y_b^*) + \hat{\rho} E \hat{D} E^\top (\zeta_a^* - \zeta_b^*) &= \mathbb{0}_m \\
\Gamma(y_a^* - y_b^*) &= \mathbb{0}_m.
\end{aligned}
\tag{19a}
\tag{19b}
$$

Note that $\Gamma E = E$. Left multiplying (19a) with $(E \hat{D} E^\top (\zeta_a^* - \zeta_b^*))^\top \Gamma$ yields $(E \hat{D} E^\top (\zeta_a^* - \zeta_b^*))^\top \hat{\rho} E \hat{D} E^\top (\zeta_a^* - \zeta_b^*) = 0$, which holds if and only if $E \hat{D} E^\top (\zeta_a^* - \zeta_b^*) = \mathbb{0}_m$ since $\hat{\rho} \succ 0$.

Hence, $(\boldsymbol{y}_a^* - \boldsymbol{y}_b^*) = -\hat{\boldsymbol{\rho}}\boldsymbol{E}\hat{\boldsymbol{D}}\boldsymbol{E}^\top(\boldsymbol{\zeta}_a^* - \boldsymbol{\zeta}_b^*) = \mathbb{0}_m$ and therefore $\boldsymbol{y}_a^* = \boldsymbol{y}_b^*$. Note that $ker(\boldsymbol{E}\hat{\boldsymbol{D}}\boldsymbol{E}^\top) = Im(\mathbb{1}_m)$ and $Im(\Gamma) \perp Im(\mathbb{1}_m)$, thus $(\boldsymbol{\zeta}_a^* - \boldsymbol{\zeta}_b^*) \perp Im(\mathbb{1}_m)$. Hence, $\boldsymbol{E}\hat{\boldsymbol{D}}\boldsymbol{E}^\top(\boldsymbol{\zeta}_a^* - \boldsymbol{\zeta}_b^*) = \mathbb{0}_m$ if and only if $\boldsymbol{\zeta}_a^* = \boldsymbol{\zeta}_b^*$. Theorefore, for every equilibrium $\boldsymbol{u}^* \in \mathbb{R}^m$, there is a unique $\boldsymbol{x}^* = (\boldsymbol{\zeta}^*, \boldsymbol{y}^*) \in \mathbb{R}^n$ such that $\boldsymbol{f}(\boldsymbol{x}^*, \boldsymbol{u}^*) = \mathbb{0}_n$.

Let the storage function be $S(\boldsymbol{x}, \boldsymbol{x}^*) = \frac{1}{2}(\boldsymbol{y} - \boldsymbol{y}^*)^\top \hat{\boldsymbol{\kappa}}^{-1}\hat{\boldsymbol{\rho}}^{-1}(\boldsymbol{y} - \boldsymbol{y}^*) + \frac{1}{2}\boldsymbol{\zeta}^\top \boldsymbol{E}\hat{\boldsymbol{D}}\boldsymbol{E}^\top\boldsymbol{\zeta}$. Then

$$
\begin{aligned}
\dot{S}(\boldsymbol{x}, \boldsymbol{x}^*) &= (\boldsymbol{y} - \boldsymbol{y}^*)^\top \hat{\boldsymbol{\rho}}^{-1}\hat{\boldsymbol{\kappa}}^{-1}\dot{\boldsymbol{y}} + \boldsymbol{\zeta}^\top \boldsymbol{E}\hat{\boldsymbol{D}}\boldsymbol{E}^\top\dot{\boldsymbol{\zeta}} \\
&= (\boldsymbol{y} - \boldsymbol{y}^*)^\top \hat{\boldsymbol{\rho}}^{-1}\left(-(\boldsymbol{y} - \boldsymbol{\lambda}^0) + \hat{\boldsymbol{\rho}}\left(\boldsymbol{u} - \boldsymbol{E}\hat{\boldsymbol{D}}\boldsymbol{E}^\top\boldsymbol{\zeta}\right)\right) + \boldsymbol{\zeta}^\top \boldsymbol{E}\hat{\boldsymbol{D}}\boldsymbol{E}^\top\boldsymbol{\Gamma}\boldsymbol{y} \\
&\overset{①}{=} -\hat{\boldsymbol{\rho}}^{-1}\|\boldsymbol{y} - \boldsymbol{y}^*\|_2^2 + (\boldsymbol{y} - \boldsymbol{y}^*)^\top(\boldsymbol{u} - \boldsymbol{u}^*) \\
&\overset{②}{\leq} -(\min_i \rho_i^{-1})\|\boldsymbol{y} - \boldsymbol{y}^*\|_2^2 + (\boldsymbol{y} - \boldsymbol{y}^*)^\top(\boldsymbol{u} - \boldsymbol{u}^*)
\end{aligned}
$$

where ① follows from $\left(-(\boldsymbol{y}^* - \boldsymbol{\lambda}^0) + \hat{\boldsymbol{\rho}}\left(\boldsymbol{u}^* - \boldsymbol{E}\hat{\boldsymbol{D}}\boldsymbol{E}^\top\boldsymbol{\zeta}^*\right)\right) = \mathbb{0}_m$ and $\boldsymbol{E}^\top\boldsymbol{y}^* = \boldsymbol{E}^\top\boldsymbol{\Gamma}\boldsymbol{y}^* = \mathbb{0}_e$ by definition of equilibrium. The inequality ② follows from $\hat{\boldsymbol{\rho}} > 0$. Therefore, the dynamics in (17) satisfy conditions in Assumption 2.

### B.1.3  Simulation and Visualization

**Simulation and training setup**   We adopt the model setup and parameters in [3, 23]. The number of vehicles is $m = 20$. The sensitivity parameter is $\kappa_i = 1$ for all vehicles. The parameters $\lambda_i^0$ and $\rho_i$ are randomly generated by $\lambda_i^0 \sim \texttt{uniform}[5, 6]$ and $\rho_i \sim \texttt{uniform}[1, 2]$, respectively. We train for 400 epochs, where each epoch trains with 300 trajectories with initial velocities $y_i(0) \sim \texttt{uniform}[5, 6]$ and initial integral variable $s_i(0) = 0$. The stepsize in time is set as $\Delta t = 0.02s$ and for $K = 300$ steps in a trajectory (Namely, each trajectory evolves 6s).

We implement control law in $\boldsymbol{u}$ to realize a specific output agreement at $\bar{\boldsymbol{y}}$ and reduce the transient cost. The transient cost is set to be $J(\boldsymbol{y}, \boldsymbol{u}) = \sum_{k=1}^{K}\|\boldsymbol{y}(k\Delta t) - \bar{\boldsymbol{y}}\|_1 + \hat{\boldsymbol{c}}\|\boldsymbol{u}(k\Delta t)\|_2^2$, where $\hat{\boldsymbol{c}} = \text{diag}(c_1, \cdots, c_m)$ with $c_i \sim \texttt{uniform}[0.025, 0.075]$. The loss function in training is set to be the same as $J(\boldsymbol{y}, \boldsymbol{u})$, such that neural networks are optimized to reduce transient cost through training. The neural PI controller can be trained by most model-based or model-free algorithms, and we use the model-based framework in [8, 53] by embedding the system dynamic model in the computation graph shown in Figure 4 and training Neural-PI by gradient descent through $J(\boldsymbol{y}, \boldsymbol{u})$.

**Controller performances.**   We compare the performance of 1) Neural-PI: the learned structured Neural-PI controllers parametrized by (7) with three layers and 20 neurons in each hidden layer. The neural networks are updated using Adam with learning rate initializes at 0.05 and decays every 50 steps with a base of 0.7. 2) DenseNN-PI: The proportional and integral terms are parameterized by dense neural networks (5) with three layers, 20 neurons in each hidden layer, and unconstrained weights. The neural networks are updated using Adam with learning rate initializes at 0.035 and decays every 50 steps with a base of 0.7. 3) Linear-PI: linear PI control where $\boldsymbol{p}(\bar{\boldsymbol{y}} - \boldsymbol{y}) := \boldsymbol{K}_P(\bar{\boldsymbol{y}} - \boldsymbol{y})$, $\boldsymbol{r}(\boldsymbol{s}) := \boldsymbol{K}_I(\boldsymbol{s})$ with $\boldsymbol{K}_P$ and $\boldsymbol{K}_I$ being the trainable proportional and integral coefficients. The coefficients are updated using Adam with learning rate initializes at 0.03 and decays every 50 steps with a base of 0.7. All of them have no communication constraints. All of the controllers are trained using 5 random seeds. The training time is shown in Table 1.

Table 1: Training time for vehicle platoon control

| Method | Average Training time (s) | Standard Deviation (s) |
|---|---|---|
| Neural-PI | 5232.36 | 30.55 |
| DenseNN-PI | 3567.01 | 16.28 |
| Linear-PI | 1836.93 | 10.09 |

The average batch loss during epochs of training with 5 seeds is shown in Figure 7(a). All of the three methods converge, with the Neural-PI achieves the lowest cost. Figure 7(b) shows the transient

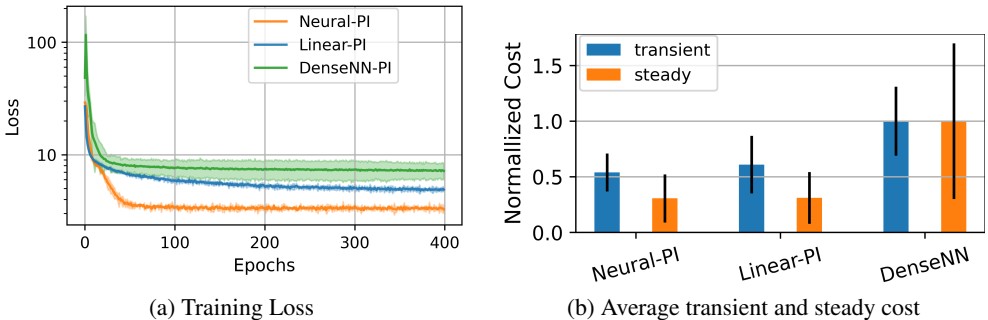

(a) Training Loss        (b) Average transient and steady cost

Figure 7: (a) The average batch loss during epochs of training with 5 seeds. All converge, with the Neural-PI achieving the lowest cost. (b) The average transient cost and steady-state cost with error bar on 100 testing trajectories starting from randomly generated initial states. Neural-PI achieves a transient cost that is much lower than others. DenseNN without structured design has both high costs in transient and steady-state performances.

and steady-state cost on 100 testing trajectories starting from randomly generated initial states. The steady-state cost is $C(\boldsymbol{y}, \boldsymbol{u}) = ||\boldsymbol{y}(15) - \bar{\boldsymbol{y}}||_1 + \hat{\boldsymbol{c}}||\boldsymbol{u}(15)||_2^2$, where we use the variables at the time $t = 15s$ since the dynamics approximately enter the steady state after $t = 15s$ as we will show later in simulation. Neural-PI and Linear-PI have the lowest steady-state cost, and the output reaches $\bar{\boldsymbol{y}}$ as guaranteed by Theorem 2. Neural-PI also achieves a transient cost that is much lower than others. By contrast, DenseNN-PI without structured design has both high costs in transient and steady-state performances.

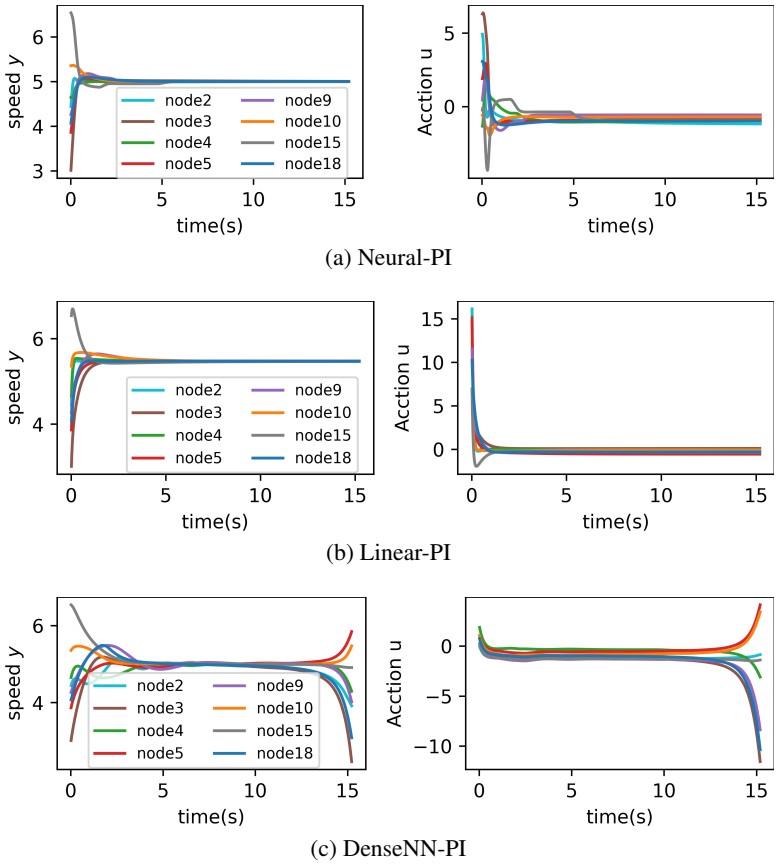

(a) Neural-PI

(b) Linear-PI

(c) DenseNN-PI

Figure 8: Dynamics of velocity $\boldsymbol{y}$ and control action $\boldsymbol{u}$ with $\bar{y} = 5$m/s. (a) Neural-PI stabilizes to $\bar{y}$ quickly. (b) Linear-PI achieves output tracking with high control effort. (c) DenseNN leads to unstable behavior.

Given $\bar{y} = 5$m/s, Figure 8 shows the dynamics of velocity $\boldsymbol{y}$ and control action $\boldsymbol{u}$ on 8 nodes under the three methods. As guaranteed by Theorem 2, Neural-PI in Figure 8(a) and Linear-PI in Figure 8(b) reaches the required speed $\bar{y} = 5$m/s. However, Linear-PI has slower convergence and much larger control efforts compared with Neural-PI. Even though DenseNN-PI achieves finite loss both in training and testing, Figure 8(c) actually exhibits unstable behaviors. In particular, DenseNN-PI appears to be stable until about 10s, but states blows up quickly after that. Therefore, enforcing stabilizing structures is essential.

## B.2 Power systems frequency control

### B.2.1 Problem statement

The second experiment is the power system frequency control on the IEEE 39-bus New England system [52] shown in Figure 9, where $\boldsymbol{u} \in \mathbb{R}^m$ is the control signal to adjust the power injection from generators and the output $\boldsymbol{y} \in \mathbb{R}^m$ is the rotating speed (i.e., frequency) of generators. The objective is to stabilize generators at the required frequency $\bar{y} = 60$Hz at the steady state while minimizing the transient control cost. The state is $\boldsymbol{x} = (\boldsymbol{\delta}, \boldsymbol{y})$, where $\boldsymbol{\delta} \in \mathbb{R}^m$ is the rotating angle of generators in the center-of-inertia coordinates with $\boldsymbol{\delta}(0) \perp Im(\mathbb{1}_m)$ [54]. The model of power systems reflects the transmission of electricity from generators to loads through power transmission lines and is represented as follows:

$$
\begin{aligned}
\dot{\boldsymbol{\delta}} &= \boldsymbol{\Gamma} \boldsymbol{y}, \\
\hat{\boldsymbol{M}} \dot{\boldsymbol{y}} &= -\hat{\boldsymbol{D}}(\boldsymbol{y} - \bar{\boldsymbol{y}}) - \boldsymbol{d} + \boldsymbol{u} - \boldsymbol{E} \hat{\boldsymbol{b}} \sin(\boldsymbol{E}^\top \boldsymbol{\delta}),
\end{aligned}
\tag{20}
$$

where $\hat{\boldsymbol{M}} = \text{diag}(M_1, \cdots, M_m)$, $\hat{\boldsymbol{D}} = \text{diag}(D_1, \cdots, D_m)$ with $M_j > 0$ and $D_j > 0$ being the inertia and damping constant of generator $j$, respectively. The vector $\boldsymbol{d}$ is the net load of the system. The matrix $\boldsymbol{E} \in \mathbb{R}^{m \times e}$ is the incidence matrix corresponding to the topology of the power network with $e$ transmission lines and satisfying $ker(\boldsymbol{E}^\top) = Im(\mathbb{1}_m)$. The matrix $\boldsymbol{\Gamma} := \boldsymbol{I}_m - \frac{1}{m} \mathbb{1}_m \mathbb{1}_m^\top$ extracts the relative rotating speed of generators by $\boldsymbol{\Gamma} \boldsymbol{y}$. The diagonal matrix $\hat{\boldsymbol{b}} = \text{diag}(b_1, \cdots, b_e) \in \mathbb{R}^{e \times e}$ with $b_j > 0$ being the susceptance of the $j$-th transmission line.

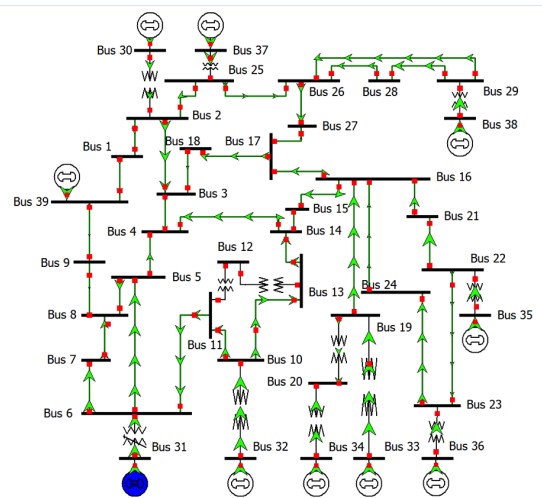

Figure 9: IEEE 39-bus test system [52]

We adopt a common assumption in literature that the power system operates with $\boldsymbol{\delta}$ satisfying $\mathcal{H} = \left\{ \boldsymbol{\delta} | [\boldsymbol{E}^\top \boldsymbol{\delta}]_j \in (-\pi/2, \pi/2) \, \forall j = 1, \cdots, e \right\}$, where $[\boldsymbol{E}^\top \boldsymbol{\delta}]_j$ is the angle difference between the generators in head and tail of the $j$-th transmission line [55–57]. This range is sufficiently large to include almost all practical scenarios [55–57].

### B.2.2 Verification of Assumption 2

At the equilibrium, the right side of (17) equals zero gives

$$-\hat{\boldsymbol{D}}(\boldsymbol{y}^* - \bar{\boldsymbol{y}}) - \boldsymbol{d} + \boldsymbol{u}^* - \boldsymbol{E}\hat{\boldsymbol{b}}\sin(\boldsymbol{E}^\top \boldsymbol{\delta}^*) = \mathbb{0}_m \text{ and } \boldsymbol{\Gamma}\boldsymbol{y}^* = \mathbb{0}_m. \tag{21}$$

We start by verifying the uniqueness of $\boldsymbol{x}^*$ for any $\boldsymbol{u}^* \in \mathcal{U}$ where (21) has a feasible solution such that $\boldsymbol{\delta} \in \mathcal{H}$. For a given $\boldsymbol{u}^* \in \mathcal{U}$, suppose there exists $\boldsymbol{x}_a^* = (\boldsymbol{\delta}_a^*, \boldsymbol{y}_a^*)$ and $\boldsymbol{x}_b^* = (\boldsymbol{\delta}_b^*, \boldsymbol{y}_b^*)$, $\boldsymbol{x}_a^* \neq \boldsymbol{x}_b^*$ such that (21) holds. Plugging in (21) gives

$$\hat{\boldsymbol{D}}(\boldsymbol{y}_a^* - \boldsymbol{y}_b^*) + \boldsymbol{E}\hat{\boldsymbol{b}}\left(\sin(\boldsymbol{E}^\top \boldsymbol{\delta}_a^*) - \sin(\boldsymbol{E}^\top \boldsymbol{\delta}_b^*)\right) = \mathbb{0}_m \tag{22a}$$

$$\boldsymbol{\Gamma}(\boldsymbol{y}_a^* - \boldsymbol{y}_b^*) = \mathbb{0}_m. \tag{22b}$$

Note that $\boldsymbol{\Gamma}\boldsymbol{E} = \boldsymbol{E}$. Left multiplying (22a) with $(\boldsymbol{E}\hat{\boldsymbol{b}}\left(\sin(\boldsymbol{E}^\top \boldsymbol{\delta}_a^*) - \sin(\boldsymbol{E}^\top \boldsymbol{\delta}_b^*)\right))^\top \boldsymbol{\Gamma}\hat{\boldsymbol{D}}^{-1}$ yields $(\boldsymbol{E}\hat{\boldsymbol{b}}\left(\sin(\boldsymbol{E}^\top \boldsymbol{\delta}_a^*) - \sin(\boldsymbol{E}^\top \boldsymbol{\delta}_b^*)\right))^\top \hat{\boldsymbol{D}}^{-1}(\boldsymbol{E}\hat{\boldsymbol{b}}\left(\sin(\boldsymbol{E}^\top \boldsymbol{\delta}_a^*) - \sin(\boldsymbol{E}^\top \boldsymbol{\delta}_b^*)\right)) = 0$, which holds if and only if $(\boldsymbol{E}\hat{\boldsymbol{b}}\left(\sin(\boldsymbol{E}^\top \boldsymbol{\delta}_a^*) - \sin(\boldsymbol{E}^\top \boldsymbol{\delta}_b^*)\right)) = \mathbb{0}_m$ since $\hat{\boldsymbol{D}}^{-1} \succ 0$. Plugging in (22a) gives $\hat{\boldsymbol{D}}(\boldsymbol{y}_a^* - \boldsymbol{y}_b^*) = \mathbb{0}_m$, which holds if and only if $\boldsymbol{y}_a^* = \boldsymbol{y}_b^*$ since $D_i > 0$ for all $i = 1, \cdots, m$.

Left multiplying $(\boldsymbol{E}\hat{\boldsymbol{b}}\left(\sin(\boldsymbol{E}^\top \boldsymbol{\delta}_a^*) - \sin(\boldsymbol{E}^\top \boldsymbol{\delta}_b^*)\right)) = \mathbb{0}_m$ with $(\boldsymbol{\delta}_a^* - \boldsymbol{\delta}_b^*))^\top$ yields

$$\begin{aligned}
0 &= \left(\boldsymbol{E}^\top \boldsymbol{\delta}_a^* - \boldsymbol{E}^\top \boldsymbol{\delta}_b^*\right)^\top \hat{\boldsymbol{b}}\left(\sin(\boldsymbol{E}^\top \boldsymbol{\delta}_a^*) - \sin(\boldsymbol{E}^\top \boldsymbol{\delta}_b^*)\right) \\
&= \sum_{j=1}^{e} b_j \left([\boldsymbol{E}^\top \boldsymbol{\delta}_a^*]_j - [\boldsymbol{E}^\top \boldsymbol{\delta}_b^*]_j\right)\left(\sin([\boldsymbol{E}^\top \boldsymbol{\delta}_a^*]_j) - \sin([\boldsymbol{E}^\top \boldsymbol{\delta}_b^*]_j)\right).
\end{aligned} \tag{23}$$

Since $b_j > 0$ and $\sin(\cdot)$ is strictly increasing in $(-\pi/2, \pi/2)$, (23) holds if and only if $\boldsymbol{E}^\top(\boldsymbol{\delta}_a^* - \boldsymbol{\delta}_a^*) = \mathbb{0}_e$. Note that $Im(\boldsymbol{\Gamma}) \perp Im(\mathbb{1}_m)$, thus $(\boldsymbol{\delta}_a^* - \boldsymbol{\delta}_b^*) \perp Im(\mathbb{1}_m)$. Hence, (23) holds if and only if $\boldsymbol{\delta}_a^* = \boldsymbol{\delta}_b^*$. Therefore, for every equilibrium $\boldsymbol{u}^* \in \mathcal{U}$, there is a unique $\boldsymbol{x}^* = (\boldsymbol{\delta}^*, \boldsymbol{y}^*) \in \mathbb{R}^n$ such that $\boldsymbol{f}(\boldsymbol{x}^*, \boldsymbol{u}^*) = \mathbb{0}_n$.

Let the storage function be $S(\boldsymbol{x}, \boldsymbol{x}^*) = \frac{1}{2}(\boldsymbol{y} - \boldsymbol{y}^*)^\top \hat{\boldsymbol{M}}(\boldsymbol{y} - \boldsymbol{y}^*) - \mathbb{1}_e^\top \hat{\boldsymbol{b}}(\cos(\boldsymbol{E}^\top \boldsymbol{\delta}) - \cos(\boldsymbol{E}^\top \boldsymbol{\delta}^*)) - (\boldsymbol{E}\hat{\boldsymbol{b}}\sin(\boldsymbol{E}^\top \boldsymbol{\delta}^*))^\top(\boldsymbol{\delta} - \boldsymbol{\delta}^*))$. Note that $-\mathbb{1}_e^\top \hat{\boldsymbol{b}}(\cos(\boldsymbol{E}^\top \boldsymbol{\delta})$ is strictly convex in $\mathcal{H}$, thus the Bregman distance $-\mathbb{1}_e^\top \hat{\boldsymbol{b}}(\cos(\boldsymbol{E}^\top \boldsymbol{\delta}) - \cos(\boldsymbol{E}^\top \boldsymbol{\delta}^*)) - (\boldsymbol{E}\hat{\boldsymbol{b}}\sin(\boldsymbol{E}^\top \boldsymbol{\delta}^*))^\top(\boldsymbol{\delta} - \boldsymbol{\delta}^*) \geq 0$ with equality holds only when $\boldsymbol{\delta} = \boldsymbol{\delta}^*$.

The time derivative is

$$\begin{aligned}
\dot{S}(\boldsymbol{x}, \boldsymbol{x}^*) &= (\boldsymbol{y} - \boldsymbol{y}^*)^\top \hat{\boldsymbol{M}}\dot{\boldsymbol{y}} + (\boldsymbol{E}\hat{\boldsymbol{b}}\sin(\boldsymbol{E}^\top \boldsymbol{\delta}) - \boldsymbol{E}\hat{\boldsymbol{b}}\sin(\boldsymbol{E}^\top \boldsymbol{\delta}^*))^\top \dot{\boldsymbol{\delta}} \\
&= (\boldsymbol{y} - \boldsymbol{y}^*)^\top(-\hat{\boldsymbol{D}}(\boldsymbol{y} - \bar{\boldsymbol{y}}) - \boldsymbol{d} + \boldsymbol{u} - \boldsymbol{E}\hat{\boldsymbol{b}}\sin(\boldsymbol{E}^\top \boldsymbol{\delta})) \\
&\quad + (\boldsymbol{E}\hat{\boldsymbol{b}}\sin(\boldsymbol{E}^\top \boldsymbol{\delta}) - \boldsymbol{E}\hat{\boldsymbol{b}}\sin(\boldsymbol{E}^\top \boldsymbol{\delta}^*))^\top \boldsymbol{\Gamma}\boldsymbol{y} \\
&\quad - (\boldsymbol{y} - \boldsymbol{y}^*)^\top \underbrace{(-\hat{\boldsymbol{D}}(\boldsymbol{y}^* - \bar{\boldsymbol{y}}) - \boldsymbol{d} + \boldsymbol{u}^* - \boldsymbol{E}\hat{\boldsymbol{b}}\sin(\boldsymbol{E}^\top \boldsymbol{\delta}^*))}_{=\mathbb{0}_m} \\
&\overset{①}{=} -(\boldsymbol{y} - \boldsymbol{y}^*)^\top \hat{\boldsymbol{D}}(\boldsymbol{y} - \boldsymbol{y}^*) + (\boldsymbol{y} - \boldsymbol{y}^*)^\top(\boldsymbol{u} - \boldsymbol{u}^*) \\
&\quad - (\boldsymbol{y}^*)^\top(\boldsymbol{E}\hat{\boldsymbol{b}}\sin(\boldsymbol{E}^\top \boldsymbol{\delta}) - \boldsymbol{E}\hat{\boldsymbol{b}}\sin(\boldsymbol{E}^\top \boldsymbol{\delta}^*)) \\
&\overset{②}{\leq} -(\min_i D_i)\|\boldsymbol{y} - \boldsymbol{y}^*\|_2^2 + (\boldsymbol{y} - \boldsymbol{y}^*)^\top(\boldsymbol{u} - \boldsymbol{u}^*)
\end{aligned}$$

where ① follows from $(-\hat{\boldsymbol{D}}(\boldsymbol{y}^* - \bar{\boldsymbol{y}}) - \boldsymbol{d} + \boldsymbol{u}^* - \boldsymbol{E}\hat{\boldsymbol{b}}\sin(\boldsymbol{E}^\top \boldsymbol{\delta}^*)) = \mathbb{0}_m$ by definition of equilibrium. The relation ② follows from $\boldsymbol{E}^\top \boldsymbol{y}^* = \boldsymbol{E}^\top \boldsymbol{\Gamma}\boldsymbol{y}^* = \mathbb{0}_e$ and $D_i > 0$ for all $i = 1, \cdots, m$. Therefore, the dynamics (20) of the power system frequency control satisfies conditions in Assumption 2.

### B.2.3 Simulation and Visualization

**Simulation Setup**   We conduct experiments on the IEEE New England 10-machine 39-bus (NE39) power network with parameters given in [52, 8]. We implement control law for power output $\boldsymbol{u}$ of generators to realize the track of frequency at 60Hz and reduce the power generation cost. The state $\boldsymbol{\delta}$ is initialized as the solution of power flow at the nominal frequency and $\boldsymbol{s}$ is initialized as 0.

The number of epochs and batch size are 400 and 300, respectively. The step-size in time is set as $\Delta t = 0.01s$ and the number of time stages in a trajectory for training is $K = 400$.

Apart from the accumulated frequency deviation, an important metric for the frequency control problem is the maximum frequency deviation (also known as the frequency nadir) after a disturbance [8]. Hence, the transient cost is set to be $J(\boldsymbol{y}, \boldsymbol{u}) = \sum_{i=1}^{n} \big( \max_{k=1,\cdots,K} |y_i(k\Delta t) - \bar{y}| + 0.05 \sum_{k=1}^{K} |y_i(k\Delta t) - \bar{y}| + 0.005 \sum_{k=1}^{K} (u_i(k\Delta t))^2 \big)$. The loss function in training is $J(\boldsymbol{y}, \boldsymbol{u})$, such that neural networks are optimized to reduce transient cost. The neural PI controller can be trained by most model-based or model-free algorithms, and we use the model-based framework in [8, 53] by embedding the system dynamic model in the computation graph shown in Figure 4 and training Neural-PI by gradient descent through $J(\boldsymbol{y}, \boldsymbol{u})$.

Two major goals of this experiment is

1) *Verifies the robustness of the controller under parameter changes*. Note that the load $\boldsymbol{d}$ is a parameter in the dynamics (20). In particular, power system operator emphasizes on the ability of the system to withstand a big disturbance such as a step load change. To this end, we train and test controllers by randomly picking at most three generators to have a step load change uniformly distributed in `uniform`$[-1, 1]$ p.u., where 1p.u.=100 MW is the base unit of power for the IEEE-NE39 test system.
2) *Verifies the performances under communication constraints*. Most systems do not have fully connected real-time communication capabilities, so the controller needs to respect the communication constraints and we show the flexibility of Neural-PI control under different communication structures.

**Controller Performances.** We compare the performance of Neural-PI controller where 1) all the nodes can communicate 2) half of the nodes can communicate and 3) none of the nodes can communicate (thus the controller is decentralized), respectively. All neural-PI controllers are parameterized by (7) and (8) where each SCNN has three layers and 20 neurons in each hidden layer. The neural networks are updated using Adam with the learning rate initializes at 0.05 and decays every 50 steps with a base of 0.7. We compare against the following two benchmarks where all the nodes can communicate: 4) DenseNN-PI-Full: Dense neural networks (5) with three layers, 20 neurons in each hidden layer, and unconstrained weights. The neural networks are updated using Adam with a learning rate initializes at 0.01 and decays every 50 steps with a base of 0.7. Note that DenseNN needs such a small learning rate to let the training converge, the reason is that DenseNN may lead to unstable behaviors that we will see later. 5) Linear-PI-Full: linear PI control where $\boldsymbol{p}(\bar{\boldsymbol{y}} - \boldsymbol{y}) := \boldsymbol{K}_P(\bar{\boldsymbol{y}} - \boldsymbol{y})$, $\boldsymbol{r}(\boldsymbol{s}) := \boldsymbol{K}_I(\boldsymbol{s})$ with $\boldsymbol{K}_P$ and $\boldsymbol{K}_I$ being the trainable proportional and integral coefficients. The coefficients are updated using Adam with the learning rate initializes at 0.08 and decays every 50 steps with a base of 0.7. All of the controllers are trained using 5 random seeds. The training time is shown in Table 2.

Table 2: Training time for power system frequency control

| Method | Average Training time (s) | Standard Deviation (s) |
| --- | --- | --- |
| Neural-PI-Full | 4373.52 | 64.58 |
| Neural-PI-Half | 8034.92 | 115.26 |
| Neural-PI-Dec | 23549.34 | 300.95 |
| DenseNN-PI-Full | 2193.84 | 21.22 |
| Linear-PI-Full | 981.65 | 11.19 |

The average batch loss during epochs of training with 5 seeds is shown in Figure 10(a). All converge, with the Neural-PI-Full achieving the lowest cost. Figure 10(b) shows the average transient cost and steady-state cost with error bar on 100 testing trajectories subject to random step load changes. The steady-state cost is $C(\boldsymbol{y}, \boldsymbol{u}) = 0.05||\boldsymbol{y}(15) - \bar{\boldsymbol{y}}||_1 + 0.005||\boldsymbol{u}(15)||_2^2$, where we use the variables at the time $t = 15s$ since the dynamics approximately enter the steady state after $t = 15s$ as we will show later in simulation. Neural-PI-Full achieves the lowest transient and steady-state cost. Notably, the steady-state cost significantly decreases with increased communication capability. The reason is that communication serves to better allocated control efforts such that they can maintain output

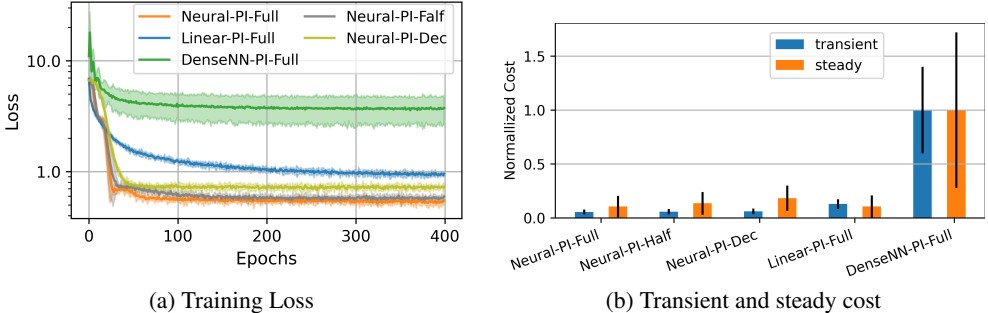

(a) Training Loss                                      (b) Transient and steady cost

Figure 10: (a) Average batch loss during epochs of training with 5 seeds. All converge, with the Neural-PI achieving the lowest cost. (b)The average transient cost and steady-state cost with error bar on 100 testing trajectories subject to random step load changes. Neural-PI achieves a transient cost that is much lower than others. The steady-state cost significantly decreases with increased communication capability. DenseNN without structured design has both high costs in transient and steady-state performances.

Table 3: The average transient cost on 100 testing trajectories starting from randomly generated initial states

| Number of training trajectories | **Neural-PI** | Linear-PI | DenseNN |
| --- | --- | --- | --- |
| 5 | **0.1328** | 0.1915 | 1.0 |
| 10 | **0.1300** | 0.1865 | 0.9833 |
| 50 | **0.1257** | 0.1838 | 0.9624 |
| 100 | **0.1234** | 0.1816 | 0.9214 |
| 300 | **0.1233** | 0.1815 | 0.5347 |

tracking with smaller control costs. Again, DenseNN without structured design has high costs both in transient and in steady state.

With a step load change at 0.5s, Figure 11 shows the dynamics of frequency $y$ and control action $u$ on 7 nodes under the five methods. Again, DenseNN-PI-Full in Figure 11(e) exhibits unstable behavior with large oscillations. As guaranteed by Theorem 2, Neural-PI in Figure 11(a-c) reaches the required frequency $\bar{y} = 60$Hz, but the speed of convergence is lower for reduced communication capabilities. Hence, the guarantees provided by the structured Neural-PI controllers are robust to parameter changes and communication constraints, which have significant practical importance.

**Performance with different numbers of training trajectories.** Table 3 compares the transient cost attained by different controllers trained with different numbers of trajectories. For both Linear-PI and Neural-PI, training with 5 trajectories for each epoch has already achieved a similar cost as training with 300 trajectories. By contrast, unstructured DenseNN requires a much larger amount of training data to reduce transient costs on testing trajectories. Therefore, the stabilizing structure significantly reduces the requirement for the number of samples to learn well.

**The impact of disturbances and noises.** The satisfaction of the Lyapunov condition is robust to disturbances in the system parameters and does not need to know how large the disturbances are, as shown in the proof Theorem 2 and Remark 2. Therefore, if there is a sudden change in the load levels, the proposed controller design still stabilizes the system and tracks the required frequency at 60Hz. In Figure 12(a), we demonstrate the system dynamics after two disturbances in load. In Figure 12(b)-(c), we add noises in both data measurement and dynamics with the signal-to-noise ratio being 5 dB (much larger than typical measurement noises). The results show that the systems are input-to-state stable, i.e., that bounded noise will lead to bounded states. Incorporating noise in rigorous theoretical analysis is an important future direction for us.

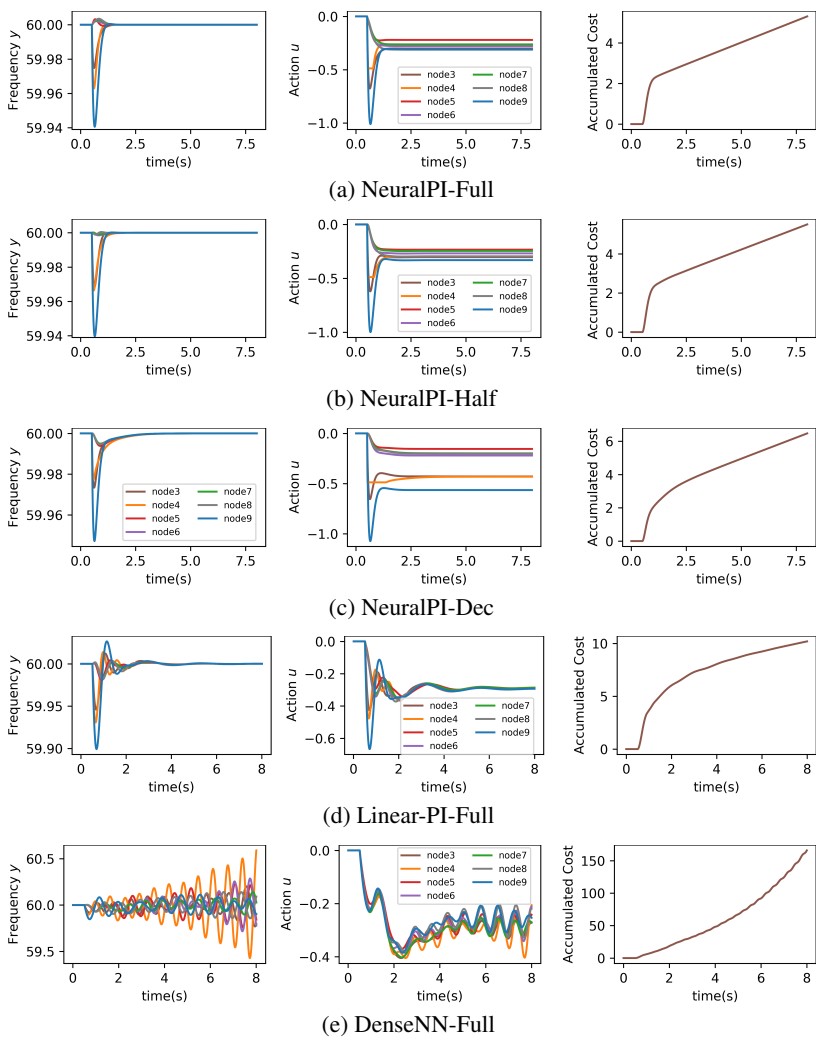

Figure 11: Dynamics of frequency $y$, control action $u$ and accumulated cost on 7 nodes with $\bar{y} = 60$Hz and a step load change at 0.5s. (a) Neural-PI when all nodes can communicate (b) Neural-PI when half of nodes can communicate, (c) Neural-PI when none nodes can communicate. The control with different communication capability all stabilize the system to the required $\bar{y} = 60$Hz. (d) Linear-PI-Full is stable but has slower convergence. (e) DenseNN-PI-Full leads to large frequency deviations and oscillations.

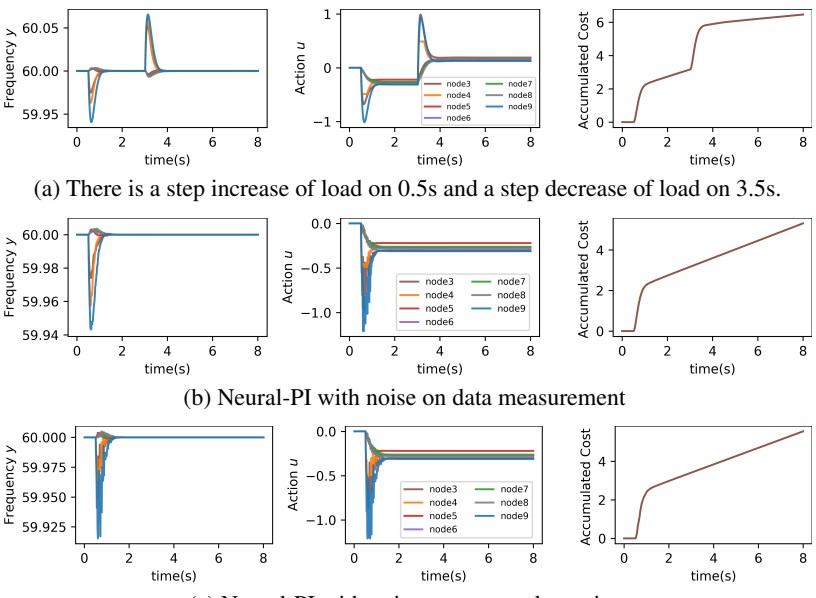

Figure 12: Frequency restoration to 60Hz after the disturbances/noises are all maintained.

