# OpenReview forum: "Structured Neural-PI Control with End-to-End Stability and Output Tracking Guarantees"
_NeurIPS.cc/2023/Conference — NeurIPS 2023 poster_

### Official Review · Reviewer_nsH5 · 2023-07-01

**Soundness:** 2 fair
**Presentation:** 2 fair
**Contribution:** 2 fair
**Rating:** 4
**Confidence:** 4

**Summary:**

This paper considers the problem of learning feedback controllers for multi-input, multi-output (MIMO) dynamical systems. At the core of the paper, a PI-like nonlinear control structure is proposed. Specifically, the controller consists of two NN, one which receives the set-point error (denoted as "P-part") and another one that receives its integral ("I-part") as an input.  For the NNs, a special structure is proposed that ensures strict convexity of the NN. This structure serves as the basis for proving stability and zero steady-state error of the closed loop system employing this controller, irrespective of the parameters used in the NN.  Hence, these guarantees can be given a-priori irrespective of the training.  Training of the NN then aims at optimizing transient behavior. Finally, two numerical examples of MIMO control systems (vehicle platooning, power network) are presented.

**Strengths:**

S1) Developing NN control architectures that are guaranteed to satisfy key control properties by design (here: stability and steady-state optimality) is very relevant. The combination of PI-like structure with strictly convex NN seems novel.

S2) Leveraging passivity properties is an interesting idea for analyzing and achieving system-wide properties. This is well-known in the control community, but should also be relvant for the NeurIPS community when considering distributed learning problems.

S3) I like the idea of having a system design that is stable and comes with base performance by design and without communication, and then improving performance further when communication is available.

**Weaknesses:**

W1)  The introduction makes some imprecise or misleading statements about the state of the art as well as technical aspects.  All statements should be carefully checked and (where needed) be backed up with proper argumentation or references. Examples:

* ll. 33-34: "Currently, learning-based approaches mostly parameterize nonlinear controllers as neural networks and train them to optimize performance".  While NN are a popular function approximator in learning-based control, they are actually used in different flavors (model learning, controller parametrization, surrogate model, etc.).  Furthermore, there are many other ML techniques that are also used, such as Gaussian processes, which are particularly popular when it comes to guarantees (examples: A Lederer, J Umlauft, S Hirche, "Uniform error bounds for Gaussian process regression with application to safe control"; C Fiedler, CW Scherer, S Trimpe, "Learning-enhanced robust controller synthesis with rigorous statistical and control-theoretic guarantees")  Hence, the characterization of the area of learning-based control as described in this above sentence is too narrow in my opinion.
* ll. 47-49: "...often overlooking the steady-state behavior (cost at the equilibrium state)."  The statement is not supported by references.  In fact, I would disagree with it, as there are works that specifically focus on the long term performance of controllers or RL policies (examples: A Doerr et al, "Optimizing long-term predictions for model-based policy search"; N Lambert et al, "Learning accurate long-term dynamics for model-based reinforcement learning")
* ll. 49-51: The description of classical P- and I-control is inaccurate.  While it is correct that I-control is often used in practice to make steady-state error zero, the actual behavior depends on the controller *and* the dynamical process.  Depending on the process dynamics (e.g., if it has an integrator itself), also a P-controller may lead to zero steady-state error. Also, the statement that P-control is for the transient, but disregarding steady-state performance is inaccurate.  P-control also affects the steady-state in general.  Again, the general description of how P- and I-control are often used in practice is not wrong, but the authors should more carefully word these sentences to avoid technical inaccuracies.
* The statement of the main research question that is addressed in this paper is too general to be representative of the paper's contribution: "Can we learn nonlinear controllers that guarantee transient stability and zero steady-state error for MIMO systems?"  (ll. 55-56).  First, the authors state themselves in the next sentence that this question can't be answered for general nonlinear system.  Second, there are existing works in learning-based control that focus specifically on MIMO PI-control (see below). IMO, the research question should thus be rephrased to already clarify the scope of this work.
* ll. 67-68, "This class of functions is too restrictive to use for controller design" -- Why?  Appropriate reasoning or reference is missing to support the statement.



W2) Related work:  There is no explicit discussion of related work as a separate section/subsection, which is not a requirement of course, but could be helpful. Some references are mentioned in the introduction, but IMO the following aspects require further discussion:

* The control problem and the objectives that are stated are super general in control (stability, zero steady-state error, fast transient). Hence, there will be a lot of works in learning-based control addressing similar problems.  While the paper does mention references and related work, I do not find this discussion very structured and have trouble pin-pointing what exactly is the difference to prior art and thus contribution of this paper.  Just saying, for example, that "guaranteeing stability and output tracking" (l. 144) is not helpful in this regard as the statement is too general and not specific to the contribution at hand.  It might help to first clearly specify the problem that is addressed, and afterwards discuss the related work specifically to the problem.
* Given that the consideration of MIMO systems is a main aspect of this work, there should be a review specific to this.  Indeed, there are existing works in learning-based control that focus specifically on MIMO PI-control (e.g., A Doerr et al, "Model-based policy search for automatic tuning of multivariate PID controllers", I Carlucho, M De Paula, GG Acosta, "An adaptive deep reinforcement learning approach for MIMO PID control of mobile robots", and many other works on "self-tuning" for MIMO systems).

Overall, I do not get a clear idea how this work compares to the state of the art.



W3) Specification of controller structure: The control law u = g(y) is not precisely specified.  Concretely, is g(.) a static function, or does it itself involve states and memory (i.e., is it itself dynamic)?  This is critical, as dynamic controllers (such as combining a state estimator and state-feedback controller) are more powerful than static maps in general. Later on, it seems that static NN are used as components of g(.).  However, the controller also involves an integrator, which would not be captured by a static function g(.) and is actually a dynamic component.  If, on the other hand, dynamic output-feedback controllers are admissible, one could presumably learn much richer control behavior.  IMO, it is ok to restrict the controller structure (from what I understand, to an integrator + static map), but it needs to be clearly and correctly stated.



W4) I am wondering whether it is really justified to call the proposed structure a proportional + integral (PI) controller. In particular, the term that is called "proportional" isn't really proportional to the error for a general nonlinear map.  I can see that what the authors propose is a generalization of well-known PI-control structure.  However, I find it a bit misleading to call this a "proportional control".  In fact, p(-y+ybar) is a general nonlinear function of the tracking error.  Hence, it will in general not be proportional to the tracking error.  Please elaborate, why you call the architecture "PI", and ideally provide references if this is a common naming.



W5) I do not find the notion of "end-to-end" guarantees conclusive and well justified.  The authors use this notion to denote the fact that they can provide certain guarantees a-priori before training (by combining a certain system class (passive) with the proposed controller architecture).  As far as I know, the term "end-to-end" is not commonly used for this (if you have references where it is used in this way, please provide them).  If a controller has theoretical guarantees, these are  most of the time "by construction", so I don't understand why this is called "end-to-end".

* Minor: In the introduction, it is not yet clear what the authors mean when mentioning "end-to-end guarantees" on l. 80. It should be explained when introduced for the first time.



W6) The presentation of experiments in the main paper is not self-contained and, IMO, insufficient. There needs to be a basic description of the problem in the main paper, which can be understood without consulting the supplementary material.  But this is currently missing.  Because of this, one cannot judge how challenging the control problem actually is, and whether sophisticated nonlinear control is actually needed.  Furthermore, the verification of assumptions is critical because a main claim of the paper are the theoretical guarantees.  This should be in the main paper.  As these are essential aspects of the contribution, the experimental section isn't self contained.  If the authors cannot fit all essential components, then a conference might be the wrong format for this manuscript.



W7) Experimental results are not fully conclusive; they compare against a standard neural network, but more classical controller structures as baseline are missing in the main paper. Thus, I find the results only partially convincing for showing the benefit of the proposed Neural-PI controller.

* It remains unclear whether the chosen problems require sophisticated nonlinear control such as the proposed Neural-PI controller. First, the problem description does not convey this (see above).  Second, in the supplementary material, the authors report the results also of a standard linear PI, whose parameters have been optimized similarly to the optimization of the NNs.  Apparently, the linear PI achieves almost the same performance as the Neural-PI for vehicle platooning, and very close performance for the power grid.  The results suggests that at least for these systems, there is no significant benefit of the proposed and more involved Neural-PI structure.  Hence, the result are not conclusive regarding the proposed architecture in this regards.
* Baseline comparisons (i.e., the linear PI from the supplementary, and maybe also a model-based controller such as LQR) should be included in the main paper to judge the performance of the proposed Neural-PI.
* I find it surprising that the DenseNN cannot stabilize the vehicle platooning system and actually has some significant deviation at the end of the horizon (Fig. 5(b), 15 sec).  Such deviation should also be visible in the objective function, so I would expect that the NN (if trained enough) can accommodate for this.
* A discussion of assumptions and theoretical guarantees is missing in the main paper. In ll. 177/178, it is claimed that in the experimental section, it will be discussed how EIP (the main assumption) can be verified if the system is unknown.  While I agree that this discussion is very important with regards of the claimed contribution of this paper, I could not find this in Sec. 6.
* The above points hold for both examples (platooning, power grid).



Minor:

* l 192: As usually, p is a gain (e.g., a real number), it would be good to emphasize when introducing it that, here, it is meant as a function of the tracking error.

**Questions:**

Q1) Why is it justified to call the structure in (4) a PI-controller, especially, why is it justified to call the first component "proportional"?

Q2) What are the 2..3 papers in existing literature that are closest to your approach? And what main limitation of these does you approach address?

Q3) In the experiments, the linear PI performs rather well. Why is the more involved Neural-PI controller even needed there?

Q4) Do you have an example, where the Neural-PI controller clearly outperforms linear approaches?

Q5) Why did you decide to not include the experimental results of the linear PI in the main paper?

**Limitations:**

Limitations (such as no consideration of disturbance/noise) are mentioned briefly in the conclusion, but not discussed in detail.  I would have liked to see a discussion on: what happens when there is noise/disturbance in the sensing, the process, and/or the data; do the authors expect that their controllers can be transferred to the real world, or what hurdles do they see?

I don't see any particular potential negative societal impact.

---

> ### Author Rebuttal · Authors · 2023-08-10
>
> We appreciate the detailed comments and thank the reviewer for the time spent on this paper.
>
> **(W1) We thank the reviewers for identifying imprecise statements. We have updated them and that would be reflected in the revised paper. Below we focus on some clarifications**:
>
> - W1-2: Steady-state behavior is generally not equivalent to long-term dynamics. It is difficult to quantify how long the trajectory is enough to reach a steady state, thus enforcing steady-state output tracking by penalizing a long-term cost is challenging. Therefore we enforce steady-state tracking by construction, instead of relying on training.
> - W1-4: Note that previous works, including those suggested by the reviewer, require the controllers to be linear. Here, we study nonlinear systems with nonlinear controllers.
> - W1-5: We study monotone functions defined as functions satisfying  $(f(x) - f(x'))^T (x - x) >0 $,  a much broader class compared to the ones considered by existing works where the function needs to satisfy $f(x) \leq f(x')$ if $x \leq x'$.
>
> **(Q2) The 3 papers closest to our approach and their limitations**
>
> - [R1]   Zhou et al. Neural Lyapunov control of unknown nonlinear systems with stability guarantees, Neurips 2022.
>   - This work learns a Lyapunov function and uses it to certify stability through a satisfiability modulo theories (SMT) solver. **But it is difficult to scale to high-dimensional systems and the learned Lyapunov function only works for a single equilibrium.**
> - [R2] Cui et al.Reinforcement learning for optimal primary frequency control: A Lyapunov approach. IEEE Trans. on Power Systems, 2022.
>   - This work shows that a class of monotone controllers can provide stability guarantees for power system frequency control problems, but it **relies on a tailor-made Lyapunov function and is **limited to SISO systems**.
> - [R3] Bürger et al. Duality and network theory in passivity-based cooperative control. Automatica, 2014.
>   -  This work derives edge feedback control for networked systems with each node being equilibrium-independent passive. **It did not optimize over the monotone functions**. For example, the paper chose the function to be tanh(), but this particular function actually is far from the optimal choice in their application.
>
> **(W2) Related work and our main contributions**
>
> - **Learning-based control with stability guarantees.**  For nonlinear systems, many works add soft penalties on the violation of stability conditions in the cost function, but it is nontrivial to certify stability for \textit{all} the possible initial states in a compact set.  Learning and certifying a Lyapunov function using SMT solvers [R1] is difficult to scale to high-dimensional systems and the learned Lyapunov function only works for a single equilibrium. The approach in [R4] for control-affine systems requires that all the states are accessible by a central agent and many systems are not control affine.**Our proposed controller guarantees stability for a set of equilibria by construction, and only needs access to the outputs (not the full states).**
>
>   [R4] Lederer et al. Uniform error bounds for gaussian process regression with application to safe control. Neurips, 2019.
>
>
> - **Algorithm to tune nonlinear and MIMO PI Controllers.** Classical PID control structures are widely used, but tuning the PID parameters is tedious. Learning-based methods become popular to tune PID parameters, although still restricted to linear control gains (including the two papers suggested by the reviewer). **Our contribution is the more generalized PI control and the MIMO neural network for parameterization**.
>
>
> **(W3, W5-7) We agree and will clarify these in the revised paper.**
>
> **(W4, Q1)  The notion of proportional + integral (PI) controller.**
>
> - We call it PI, because the name can make it easier for readers to understand the structure of the controller that is a summation of a function of the proportional variable $ -y+\bar{y}$ and a function of the integral variable $s$. Note that monotone $p(  -y+\bar{y})$ covers linear functions as a special case. Our terminology is not new, see for example, [R5], where the controller is not linear in the tracking error.
>
>   [R5] Generalized proportional integral control for periodic signals under active disturbance rejection approach, ISA Transactions, 2014.
>
> **(Q3, Q4) Comparison with Neural-PI controller**
>
> - We would like to emphasize that ** the transient cost attained by Neural-PI is significantly lower than Linear-PI.**  As we show in the (G1) of global response, Neural-PI reduces the average transient cost by approximately 30%  compared to Linear-PI (please see Fig 1 in the attached PDF in the global response).
>
> **(W7-3) DenseNN in (Fig. 5(b), 15 sec) has some significant deviation, does the objective function cover this?**
>
> - Fig. 5(b) is one representative unstable case, but it does not indicate that all cases are unstable. The objective function is useful to capture the behavior of the majority of the cases, but it is not sufficient to guarantee the behavior of all possible cases. Learning can only trains the neural network through a finite number of trajectories over a finite time horizon, so it cannot guarantee the stability of trajectories from **all** possible states.
>
> **(Q5) Why did you decide to not include the experimental results of the linear PI in the main paper?**
>
> - We did not include Linear PI in the main paper because the structure of Neural-PI involves the Linear-PI as a special case, so there is no surprise that optimizing Neural-PI would work better than Linear-PI. And we did not add it to the paper because of space limitations. We agree that including Linear-PI in the main paper can demonstrate the improvement of Neural-PI. We will include the comparisons in the revised main paper.
>
> **The consideration of noises/disturbances and real-world applications**
>
> - Please refer to (G2) and (G5) of the global response and the attached pdf.

---

> > ### Comment · Reviewer_nsH5 · 2023-08-16
> >
> > I have read the rebuttal, I have no specific follow-up questions.
> >
> > In the rebuttal, the authors have well addressed some of my previous points (e.g., clarification on the contribution). Other aspects will require significant rewriting of the manuscript to an extent that (in my opinion) should require a re-review.  Nonetheless, I changed my score reject -> borderline reject, as the weaknesses are less severe overall after the rebuttal.

---

> > > ### Author Response · Authors · 2023-08-17
> > >
> > > We are grateful to the reviewer for detailed comments and for updating the score. Because we are not allowed to update the main paper and the supplementary material in this round, we briefly describe the changes we intend to make if the paper is accepted.
> > >
> > > All the clarification and revisions shown in the rebuttal will be in the revised paper, including
> > > (1) revisions and clarifications of statements in (W1);
> > > (2) adding a section of related work to emphasize the contribution of this work compared with the state of the art;
> > > (3)  clarification of the controller structure as an integrator + static map and the notion of Neural-PI;
> > > (4) clarification of the notion  "end-to-end" that is equivalent to "by construction";
> > > (5) supplementing a basic description of the problem, the verification of assumptions, and the performance of the linear PI control in the main paper;
> > > (6) supplementing the experiments related to the impact of trained trajectories, noise, and disturbances to the supplementary material;
> > > (7) adding the discussions on the limitation and applicability in real-world applications to the conclusion.
> > >
> > > If the reviewer has any more specific questions, we are happy to address them.

---

### Official Review · Reviewer_F58a · 2023-07-03

**Soundness:** 2 fair
**Presentation:** 3 good
**Contribution:** 2 fair
**Rating:** 5
**Confidence:** 3

**Summary:**

This paper proposes a method for optimal control of dynamical systems using a neural network-based controller that guarantees stability and output tracking. The key structure of the proposed neural proportional-integral (PI) controller is based on the strict monotonicity of the proportional and integral terms, which are parameterized as the gradient of a strictly convex neural network. The paper also introduces tunable activations for universal approximation capabilities and communication constraints. Experiments on traffic and power networks show significant improvements in both transient and steady-state performance. The contribution of this paper is mainly in the field of control theory and its application.

**Strengths:**

The strength of this paper is that it presents a method for optimal control using a neural network-based controller with provable stability guarantees. Using a strictly monotonic function parameterized by a strictly convex neural network can improve control performance and output tracking. Experiments on traffic and power networks show significant improvements in both transient and steady-state performance.

**Weaknesses:**

The multiple equilibria problem has not been fully addressed in real-world applications. This limitation limits the applicability of the proposed method to systems with general nonlinear systems.

**Questions:**

In real applications, how to judge the passivity of a large-scale nonlinear system is a possible problem to extend the proposed approach.

**Limitations:**

The paper does not explicitly discuss the scalability of the proposed method to larger, more complex systems. This paper does not provide a comparison with other state-of-the-art control methods, which helps to better understand the advantages and disadvantages of the proposed method.

---

> ### Author Rebuttal · Authors · 2023-08-10
>
> We are grateful to the reviewer for thoroughly reading our paper and providing encouraging
> comments. Below, we provide a point-by-point response.
>
> **The multiple equilibria problem**
>
> - The multiple equilibria come from different setpoints of output to track.  For example, a group of vehicles may need to cruise at different speeds. Once a setpoint (e.g., the required cruise speed) is given, the equilibrium of the system becomes unique. Our proposed Neural-PI control has provable guarantees of stability and zero steady-state output tracking error for different setpoints. More specifically, **the performance guarantees in Theorem 2 hold for various $\bar{y}$, and the satisfaction of the Lyapunov condition does not depend on the specifics of $f(\cdot)$**. Therefore, our proposed method work for real-world applications with a range of possible equilibria.
>
>
> **How to judge the passivity of a large-scale nonlinear system?**
>
> - We agree that judging the passivity of a large-scale nonlinear system is an important problem. The passivity analysis of large-scale nonlinear systems, especially networked systems, is an active line of research in academia. One benefit of passive analysis is that it provides convenience for a modular analysis of large-scale systems (for details, see  [R1-R3]). **If each of the subsystems is equilibrium-independent passive (EIP),  then the whole system is still EIP if it interconnects the subsystems through parallel, negative feedback, or an undirected graph [R1-R3]**.  This allows the passivity analysis extends to large-scale nonlinear systems [R1], for example, large power networks, traffic networks, internet congestion management, etc.
>
> - More specifically, such modular analysis allows us to divide a large-scale system into smaller subsystems, and then verify the EIP of each subsystem. For a wide range of systems, a storage function has been found to judge EIP [R1]. Even if a storage function has not been found,  it is also possible to learn a storage function parameterized by neural networks for a smaller-scale subsystem, and certify EIP through a satisfiability modulo theories (SMT) solver [R4]. Since there has been lots of success in learning a neural Lyapunov function for small-scale systems(e.g., [R4] and [R5]), we envision that such a modular approach provides the possibility to extend that to larger systems.
>
>   [R1] Arcak et al. Networks of dissipative systems: compositional certification of stability, performance, and safety. Springer, 2016.
>
>   [R2] Hines et al. Equilibrium-independent passivity: new definition and numerical certification. Automatica, 2011.
>
>   [R3] Simpson et al. Equilibrium-independent dissipativity with quadratic supply rates. IEEE Transactions on Automatic Control, 2018.
>
>   [R4] Chang et al. Neural Lyapunov control. Neurips 2019.
>
>   [R5] Zhou et al. Neural Lyapunov control of unknown nonlinear systems with stability guarantees, Neurips 2022.
>
>
> **The scalability of the proposed method to larger, more complex systems**
>
> - As we discussed in the previous response, passivity enables a modular analysis that can scale to large systems.  Namely, we can divide the large-scale systems into smaller subsystems, and then verify EIP of each subsystem.
>
> - Moreover, **Neural-PI control law retains all the stability guarantees of classical linear PI control, but achieves much lower transient cost**. Since classical PI control is widely utilized in real-world applications, we expect that the controllers can be transferred to more complex systems. For example, our controllers are in the process of being implemented in a real-world test system, the UCSD DER-connect testbed (none of the authors are affiliated with the testbed). Rigorous analysis of the scalability of the proposed method to larger, more complex systems is an important future direction for us.
>
>
> **Comparison with other state-of-the-art control methods**
>
> - In supplementary material, we compared the proposed method with unstructured neural networks and classical PI control. Notably, **the focus on structure is complementary to most existing learning algorithms (e.g., model-based and model-free reinforcement learning)**: by replacing unstructured neural networks with the proposed Neural-PI structure, the controllers trained by these algorithms will provide provable guarantees of stability and zero steady-state output tracking error. This is the reason why the comparison focuses on the controller structures (i.e.,  Neural-PI, DenseNN, and classical linear PI) trained by the same algorithm. To the best of our knowledge, we have not found other structured controllers to incorporate in numerical comparisons.
>
> - The key results of comparisons are summarized as follows:
>   - The proposed Neural-PI control law retains all the stability guarantees of classical linear PI control, but **reduces the average transient cost by approximately 30% compared to Linear-PI**. This is reflected in Figure 7 and Figure 10 of the supplementary material.
>   - **Neural-PI significantly reduces the number of sampled trajectories to train well compared with DenseNN**.  This is reflected in the following table that compares the transient cost attained by different controllers trained with different numbers of trajectories. For both Linear-PI and Neural-PI, training with 5 trajectories for each epoch has already achieved a similar cost as training with 300 trajectories. By contrast, unstructured DenseNN requires a much larger amount of training data to reduce transient costs on testing trajectories.
>
>
>    **Table 1: The average transient cost on 100 testing trajectories starting from randomly generated initial states**
>
>    | # of training trajectories  |**Neural-PI** | Linear-PI |DenseNN|
>    |---|---|---|---|
>    | 5 | **0.1328** | 0.1915 |  1.0 |
>    | 10 | **0.1300**| 0.1865 | 0.9833 |
>    | 50 | **0.1257** | 0.1838  | 0.9624 |
>    | 100 | **0.1234**| 0.1816 | 0.9214 |
>    | 300 | **0.1233**| 0.1815 |  0.5347 |

---

> ### Comment · Reviewer_F58a · 2023-08-19
>
> The authors' responses are reasonable. I recommend accepting the paper.

---

> > ### Author Response · Authors · 2023-08-19
> >
> > We are grateful to the reviewer for the response and encouraging comments.

---

### Official Review · Reviewer_aJSd · 2023-07-04

**Soundness:** 4 excellent
**Presentation:** 3 good
**Contribution:** 2 fair
**Rating:** 6
**Confidence:** 5

**Summary:**

This paper presents an approach to control multi-input, multi-output dynamical systems, introducing a neural network-based PI controller offering provable stability and output tracking guarantees. The neural Proportional-Integral controller is based on strict monotonicity parameterized by strictly convex neural networks (SCNN). The strict monotonicity of the network gives the convergence guarantee by construction.  Testing on traffic and power networks showed improved transient and steady-state performances compared to unstructured networks.

**Strengths:**

It is interesting to improve PI control using neural networks. The proposed framework not only keeps the structure (which is well-validated across decades) and robust tracking performance of PI controllers, but also allows optimizing the tracking performance by learning the parameters in g and r functions. The use of monotonic neural networks is also critical. The experiment results demonstrate its advantages over regular fully connected neural networks.

**Weaknesses:**

- The authors must have known that carefully tuned PI controllers already work very well. Why didn’t they compare in the experiments? It is common that controllers built upon regular fully connected layers will diverge as time progresses, but this is not a problem in simple regular PI controllers. How well the proposed method is, compared to a carefully tuned classical PI controller?

- Even if the proposed framework is better than traditional PI control in the transient phase, how can the authors guarantee its performance in situations outside the training data distribution? In real world applications where data collection can be expensive, if we only have 1000 data points, would the authors choose to train a neural network (this could be dangerous if only have limited data) or tune a classical PI controller using their extensive domain knowledge of the system dynamics (and use the 1000 data points for tuning & experiment).

- In summary, this study is interesting and novel, but its importance / practical contribution to the control field might be minor.


**Questions:**

See weaknesses.

**Limitations:**

See weaknesses.

---

> ### Author Rebuttal · Authors · 2023-08-09
>
> We are grateful to the reviewer for thoroughly reading our paper and providing encouraging comments. We agree with the reviewer that carefully tuned PI controllers can work well for a large range of applications, and the controller by design has been shown to have good stability properties.  **Classical PI controllers are the impetus for this work, where we want to maintain good theoretical properties and improve over some of the limitations (e.g., slow convergence speed and larger control efforts).** This is the reason why we don’t use a general neural network, but a specific Neural-network structure.
>
> Below, we first show that the Neural-PI control law retains all the stability guarantees of classical linear PI control, but **achieves much lower transient cost**.  Then, we provide numerical results to demonstrate that **this performance improvement is present even if there is little data to train the controllers**.
>
>
> **Comparison to Carefully Tuned PI Controller**
>
> - We actually compared the proposed Neural-PI with regular PI controllers (labeled as Linear-PI) in the supplementary material (see Figure 7 and Figure 10). But we should have made it more clear where these results are and leaving them in the appendix may have caused confusion about how well the proposed controller compares with carefully tuned classical linear-PI control law. We will add the comparison with Linear-PI in the revised main paper.
>
> - **We would like to emphasize that the transient cost attained by Neural-PI is significantly lower than Linear-PI.**  In the following, we take the experiments of power systems as an example to show the comparison, where both Neural-PI and Linear-PI are trained with the same algorithm.
>
>    For both the trained Neural-PI and Linear-PI, the table below compares the average and standard deviation of transient cost on 100 testing trajectories starting from randomly generated initial states. Thus, Neural-PI reduces the average transient cost by 32%  compared to Linear-PI.
>
>    **Table 1: The transient cost on 100 testing trajectories starting from randomly generated initial states**
>    | Metric  |**Neural-PI** | Linear-PI |
>    |---|---|---|
>    | Transient cost mean | **0.1233** | 0.1815  |
>    | Transient cost std | **0.0677**| 0.0774 |
>
>
>
>
> **Performance of Neural-PI controller under different numbers of training data**
>
> - The proposed Neural-PI structure provides provable stability and output-tracking guarantees **by construction**. Thus, **such guarantees hold for all cases even if the scenario does not exist in training**.
> - **Compared with Linear-PI, the proposed Neural-PI does not require more data to train well**. This is reflected in the following table that compares the transient cost attained by different controllers trained with different numbers of trajectories (each trajectory evolves 400 time steps). For both Linear-PI and Neural-PI, training with 5 trajectories for each epoch has already achieved a similar cost as training with 300 trajectories. By contrast, unstructured DenseNN requires a much larger amount of training data to reduce transient costs on testing trajectories. Therefore, the stabilizing structure significantly reduces the requirement for the number of samples to learn well.
>
>    **Table 2: The average transient cost on 100 testing trajectories starting from randomly generated initial states**
>
>    | # of training trajectories  |**Neural-PI** | Linear-PI |DenseNN|
>    |---|---|---|---|
>    | 5 | **0.1328** | 0.1915 |  1.0 |
>    | 10 | **0.1300**| 0.1865 | 0.9833 |
>    | 50 | **0.1257** | 0.1838  | 0.9624 |
>    | 100 | **0.1234**| 0.1816 | 0.9214 |
>    | 300 | **0.1233**| 0.1815 |  0.5347 |
>
> - Importantly, tuning a classical PI controller to optimize over a host of objective functions can be challenging even if the exact model and parameters are available. Learning-based methods provide a flexible framework to optimize PI parameters as long as the objective function is differentiable. Considering that Neural-PI trained with 5 trajectories at each epoch already achieves a transient cost much lower than Linear-PI, we envision that Neural-PI is applicable in real-world applications.
>
>
>
>
>  **The applicability of Neural-PI compared with Linear-PI**
>
> - The above results show that the Neural-PI control law **retains all the stability guarantees of classical linear PI control, but achieves much lower transient cost**. The control theoretic guarantees also significantly reduce the amount of data required to train well. In other words, the Neural-PI controller cannot do worse than a carefully tuned PI controller. However much data there is to train a Neural-PI controller, the Neural-PI controller archives a much smaller transient cost compared with classical Linear-PI. **If there is domain knowledge to tune a classical PI controller, we can take it as a warm start and train the neural network starting from there**.
>
> - In addition, **the proposed neural network construction makes it tractable to optimize nonlinear monotone functions to improve transient performances**, even when exact parameters for the system are not available.
>
> - Since classical PI control is widely utilized in real-world applications, we expect that the controllers can be transferred to real-world scenarios. For example, our controllers are in the process of being implemented in a real-world test system, the UCSD DER-connect testbed (none of the authors are affiliated with the testbed).

---

> > ### Comment · Reviewer_aJSd · 2023-08-10
> > **Thanks for the Response**
> >
> > I would like to thank the authors for their detailed response. I don't have further questions, and will change to accept.

---

> > > ### Author Response · Authors · 2023-08-15
> > >
> > > Thank you for your response and for changing the scores to accept. We are happy that we’ve addressed the questions.

---

### Official Review · Reviewer_eDxT · 2023-07-06

**Soundness:** 3 good
**Presentation:** 4 excellent
**Contribution:** 3 good
**Rating:** 7
**Confidence:** 4

**Summary:**

The paper "Structured Neural-PI Control with End-to-End Stability and Output Tracking Guarantees" presents a well-motivated and technically sound approach to address the lack of provable guarantees in neural network-based controllers. The combination of structured neural-PI control, equilibrium-independent passivity, and SCNNs with tunable softplus-β activations demonstrates promising results in terms of stability, output tracking, and improved performance.

**Strengths:**

The paper is clear, well written and accurate in the mathematical control details. The method leverages a specific kind of NN and the approach is clean and seems to hold great potential, at least to an applicable systems class.

**Weaknesses:**

The proposed approach relies on the assumption of equilibrium-independent passivity, which may limit its applicability to a specific class of physical systems. The paper should explicitly discuss the range of systems to which the approach is applicable and highlight any potential limitations or challenges in extending it to other domains.

**Questions:**

1.	In Definition 2, is it critical to demand zero tracking error? That can be augmented easily with a prefilter or an integrator. Even though the controller has an integrator, the system may be of higher type).
2.	In 3.1 it is stated that it is desired to optimize the transient response before the system settles on the tracking point. What are the specifications? Maximum overshot? Settling time? Are there minimum requirements?

---

> ### Author Rebuttal · Authors · 2023-08-10
>
> We are grateful to the reviewer for thoroughly reading our paper and providing encouraging
> comments. Below, we provide a point-by-point response.
>
>
> **The applicability and potential limitations in extending the method to other domains.**
>
> - Theoretically, this method works for systems that satisfy EIP properties, including the transportation and power systems examples studied in the paper, and many other networked systems such as robotics and communication [R1]. For EIP systems, the proposed Neural-PI control law obtains rigorous stability and zero steady-state tracking error guarantees by construction, and results in significantly lower transient cost compared to the linear PI controller and standard NN controller (as shown in the traffic and power system simulations). For extending the proposed method to other domains, potential barriers include the verification of EIP property when a storage function is difficult to be found, and provable guarantees on the robustness to noises. These are all important future directions for us.
>
>   [R1] Murat Arcak, Chris Meissen, and Andrew Packard. Networks of dissipative systems: compositional certification of stability, performance, and safety. Springer, 2016
>
>
> **Q1: In Definition 2, is it critical to demand zero tracking error? That can be augmented easily with a prefilter or an integrator. Even though the controller has an integrator, the system may be of higher type).**
>
> - We agree that zero tracking error can be augmented with a prefilter or an integrator. We will make this more clear in the revised paper.
>
>
> **Q2: What are the specifications to optimize the transient response?**
>
> - The specifications depend on the application case by case, and it generally includes reducing state deviation (e.g., maximum overshot, the accumulated deviation from the tracking point) and control cost. The minimum requirement is that the objective function is differentiable.
>
> - For example, in power system frequency control,  an important metric is the maximum frequency deviation (also known as the frequency nadir) after a disturbance. In addition, we also hope to keep the assumed frequency deviation small with lower control effort.  Then the cost function is the summation of the maximum frequency deviation, the accumulated frequency deviation, and the control effort.

---

> > ### Comment · Reviewer_eDxT · 2023-08-10
> > **Thanks for the response**
> >
> > I would like to the authors for the response. I have no further questions.

---

> > > ### Author Response · Authors · 2023-08-15
> > >
> > > Thank you for your response and for providing encouraging comments.

---

### Official Review · Reviewer_Cp4m · 2023-07-09

**Soundness:** 3 good
**Presentation:** 3 good
**Contribution:** 3 good
**Rating:** 6
**Confidence:** 4

**Summary:**

This paper propose a neural-PI controller based on the newly designed strictly convex neural network. Thanks to the EIP condition of the considered systems, the neural-PI controller naturally satisfies the stability and output tracking guarantee with the special structure of the SCNN. The efficacy of the proposed method is demonstrated in several physical models.

**Strengths:**

[1] The proposed neural-PI controller satisfies the stability guarantee by construction of the SCNN instead of adding regularization term to the loss function, which is a hugh improvement to the existing learning controller commuity.

[2] The proposed method does not require the knowledge of the equilibria of the intrisic dynamics but can still provide stability guarantee of the unknown equilibria.

[3] The experiments take communication cost into consideration, which is realistic and instructive.

[4] The presentation of the paper is good, the workflow in the figures are clear and helpful.


**Weaknesses:**

[1] The current framework is restricted in the system with EIP condition.

[2] The authors did not compare the proposed method with the existing methods such as RL, QP and MPC.

[3] The training policy is naïve, the learner should solve the controlled ODE to find the controlled trajectory and then update the parameters.


**Questions:**

[1] Since the objective functional incurs the integral of the controlled ODE, can this method scale to the high-dimensional task?

[2] Can this method outperform the existing methods (RL such as PPO or MPC) in terms of training speed or test performance?


**Limitations:**

[1] The current neural method can only handle deterministic systems with EIP condition, which cannot adapt to the real-world scenarios where noise is unavoidable.

[2] The authors did not provide numerical comparison with the existing methods.

---

> ### Author Rebuttal · Authors · 2023-08-09
>
> We are grateful to the reviewer for thoroughly reading our paper and providing encouraging comments. Below, we provide a point-by-point response.
>
>  **Can this method outperform the existing methods (RL such as PPO or MPC)?**
>
> - The Neural-PI controller can be trained by almost all the RL algorithms, including PPO and other variants. **The focus on structure is complementary to training algorithms**: by replacing unstructured neural networks with the proposed Neural-PI structure, the trained controllers will provide provable guarantees of stability and zero steady-state output tracking error. The comparisons then focus on the controller constructions (i.e.,  Neural-PI, DenseNN, and classical linear PI) trained by the same algorithm.
>
> - In addition to the performance guarantees, we show in experiments that **Neural-PI significantly reduces the number of sampled trajectories to train well compared with DenseNN**. This can also help to mitigate the burden of data collection in these RL algorithms.
>
> - In this paper, we consider the systems without real-time communication/computation capabilities. Model predictive control (MPC) typically needs full communication and performs computation in real-time, so it is not directly applicable to systems considered in this paper.
>
> **(W3 and Q1) Questions related to ODE and the training policy**
>
> - We agree that the training method we provided is simple, although it is not very different from what the reviewer suggests here. Essentially, we embed the discretized ODE in a computation graph, which is convenient to train neural networks through backpropagation. We note that other model-based or model-free methods can be used as well.
>
> - We would like to clarify that implementing the control policy in equation (4) does not require solving a controlled ODE. For the integral variable $\dot{s}=-(y-\bar{y})$, the computation of $s$ is approximated using the measurement of $s(t) \approx s(0)+\sum_{k=0}^{t} -(y(k)-\bar{y}) \Delta t$ with $\Delta t$ be the sampling time interval. This is a common practice and is widely implemented in control. Thus, the Neural-PI takes the measures of $y$ and $s$ as inputs, which is in essence no different from a generic neural network and can scale to high-dimensional tasks.
>
>
> **Adaptation to real-world scenarios where noise is unavoidable.**
>
> We thank the reviewer for bringing up this important point. Below, we first clarify that the method is robust to disturbances in the system parameters. Then, we show experimental results that the controller also works well for systems with noises both in data measurement and in the system.
> -  **The satisfaction of the Lyapunov condition is robust to disturbances in the system parameters**, as shown in the proof Theorem 2 and Remark 2. Take the power system experiment as an example, the load $d$ in the system dynamics is a parameter. If there is a sudden change in the load levels, the proposed controller design still stabilizes the system and tracks the required frequency at 60Hz. In Figure 3(a) of the attached PDF in the global response, we demonstrate the system dynamics after two disturbances in load.
>
> - For noises in the data measurements and state transition dynamics, we empirically find that **the systems are input-to-state stable, i.e., that bounded noise will lead to bounded states**.  Incorporating noise in rigorous theoretical analysis is an important future direction for us. We have performed empirical studies with noises in both data measurement and dynamics, as shown in Figure 3(b)-(c) of the attached PDF in the global response.  The signal-to-noise ratio is about 5 dB (much larger than typical measurement noises).  Figure 3 shows that the performance guarantees are robust to noise.
>
> Overall, the EIP property has been found in a large class of physical systems, including the transportation and power systems examples studied in the paper, and many other networked systems in robotics and communication [R1]. The above results show that the design can adapt to real-world scenarios where noise is unavoidable.
>
> [R1] Networks of dissipative systems: compositional certification of stability, performance, and safety. Springer, 2016
>
> **Numerical comparisons with the existing methods.**
>
> A: In supplementary material, we compared the proposed Neural-PI with unstructured neural networks (DenseNN) and classical PI control (Linear-PI). As we mentioned above, the proposed method focuses on the structure, and thus we train the controllers using the same algorithm. To the best of our knowledge, we have not found other existing structured controllers to incorporate in numerical comparisons.
>
> The results for numerical comparisons are summarized as follows:
> - The proposed Neural-PI control law retains all the stability guarantees of classical linear PI control, but **reduces the average transient cost by approximately 30% compared to Linear-PI**. This is reflected in Figure 7 and Figure 10 of the supplementary material.
>
> - **Neural-PI significantly reduces the number of sampled trajectories to train well compared with DenseNN**.  This is reflected in the following table that compares the transient cost attained by different controllers trained with different numbers of trajectories. For both Linear-PI and Neural-PI, training with 5 trajectories for each epoch has already achieved a similar cost as training with 300 trajectories. By contrast, unstructured DenseNN requires a much larger amount of training data to reduce transient costs on testing trajectories.
>
>    **Table 1: The average transient cost on 100 testing trajectories starting from randomly generated initial states**
>
>    | # of training trajectories  |**Neural-PI** | Linear-PI |DenseNN|
>    |---|---|---|---|
>    | 5 | **0.1328** | 0.1915 |  1.0 |
>    | 10 | **0.1300**| 0.1865 | 0.9833 |
>    | 50 | **0.1257** | 0.1838  | 0.9624 |
>    | 100 | **0.1234**| 0.1816 | 0.9214 |
>    | 300 | **0.1233**| 0.1815 |  0.5347 |

---

> > ### Comment · Reviewer_Cp4m · 2023-08-12
> >
> > Thank you very much for replying to my concerns.   Good jobs and I will change my scores, suggesting to accept.

---

> > > ### Author Response · Authors · 2023-08-15
> > >
> > > Thank you for your response and for changing the scores to accept. We are happy that we’ve addressed the concerns.

---

### Author Rebuttal · Authors · 2023-08-10

We thank the reviewers for the detailed comments. We notice there are some common concerns and respond to these questions in the general response below.

We have attached a **one-page PDF containing the figures and updated numerical results**. Because of the space limits, all the numerical results are demonstrated in the case of power system frequency control. Note that we are not allowed to update the main paper and supplementary material in this round, so the changes would be reflected when possible.

**(G1) The comparison with Linear-PI**

Classical PI controllers are the impetus for this work, where we want to maintain good theoretical properties and improve over some of the limitations (e.g., slow convergence speed and larger control efforts). This is the reason why we don’t use a general neural network, but a specific Neural-network structure. The key comparison results are as follows:

- The structure of Neural-PI involves the classical PI controllers (i.e., Linear-PI) as a special case, and the theoretical results in Theorem 2 show that  **Neural-PI control law retains all the stability guarantees of classical linear PI control**.

- Experimental results in the supplementary material showed that the **Neural-PI achieves much lower transient cost compared with Linear-PI**. In particular, **the performance improvement is present even if there is little data to train the controllers**.  For Neural-PI, Linear-PI, and DenseNN trained with different amounts of trajectories, the table below compares the average transient cost on testing trajectories starting from randomly generated initial states. For all the cases, Neural-PI reduces the average transient cost by approximately 30% compared to Linear-PI. Moreover, Neural-PI significantly reduces the number of sampled trajectories to train well compared with DenseNN. The comparison with the error bar and the dynamics are visualized in Figure 1 and Figure 2 of the attached PDF, respectively.

   **Table 1: The average transient cost on 100 testing trajectories starting from randomly generated initial states**

   | # of training trajectories  |**Neural-PI** | Linear-PI |DenseNN|
   |---|---|---|---|
   | 5 | **0.1328** | 0.1915 |  1.0 |
   | 10 | **0.1300**| 0.1865 | 0.9833 |
   | 50 | **0.1257** | 0.1838  | 0.9624 |
   | 100 | **0.1234**| 0.1816 | 0.9214 |
   | 300 | **0.1233**| 0.1815 |  0.5347 |





**(G2) The impact of noises/disturbances**


- The satisfaction of the Lyapunov condition is **robust to disturbances in the system parameters and does not need to know how large the disturbances** are, as shown in the proof Theorem 2 and Remark 2. Take the power system experiment as an example, the load $d$ in the system dynamics is a parameter. If there is a sudden change in the load levels, the proposed controller design still stabilizes the system and tracks the required frequency at 60Hz. In Figure 3(a) of the attached PDF in the global response, we demonstrate the system dynamics after two disturbances in load.

- For noises in the data measurements and state transition dynamics, we empirically find that **the systems are input-to-state stable, i.e., that bounded noise will lead to bounded states**.  Incorporating noise in rigorous theoretical analysis is an important future direction for us. We have performed empirical studies with noises in both data measurement and dynamics, as shown in Figure 3(b)-(c) of the attached PDF in the global response.  The signal-to-noise ratio is about 5 dB (much larger than typical measurement noises).  Figure 3 shows that the performance guarantees are robust to noise.




**(G3) Clarification of numerical comparisons**

- Although there is a vast literature on training algorithms (e.g., model-based and model-free reinforcement learning) that either train a neural network or a classical PID controller, the stability guarantees that can be achieved in the structure of neural networks has rarely been explored. Indeed, **The focus on structure is complementary to training algorithms**: by replacing unstructured neural networks with the proposed Neural-PI structure, the trained controllers will provide provable guarantees of stability and zero steady-state output tracking error.  The comparisons then focus on the structure of controllers  (i.e.,  Neural-PI, DenseNN, and classical linear PI) trained by the same algorithm.

**(G4) Summary of contributions**
- Our contribution is the Neural-PI controller that has provable guarantees of stability and zero steady-state output tracking error, and features 1)  the stability guarantee holds for a set of equilibria, 2) multi input multi output (MIMO) monotone neural network architecture and universal approximation theorem, 3) capability to respect the communication constraints.

**(G5) The applicability to the real world**

- The above results show that the **Neural-PI control law retains all the stability guarantees of classical linear PI control, but achieves much lower transient cost**. The control theoretic guarantees also significantly reduce the amount of data required to train well. Since classical PI control is widely utilized in real-world applications, we expect that the controllers can be transferred to real-world scenarios. For example, our controllers are in the process of being implemented in a real-world test system, the UCSD DER-connect testbed (none of the authors are affiliated with the testbed).
- Potential barriers include the verification of EIP when a storage function is difficult to be found and provable guarantees on the robustness to noises. These are all important future directions for us.

---

### Decision · Program_Chairs · 2023-09-21

**Decision:**

Accept (poster)

**Comment:**

The paper proposes a generalization of the classic proportional + integral (PI) control law to be parametrized by neural networks. It is empirically observed that performance is better than the classical PI method on several example systems. The proposed structure also comes with certain correct-by-construction guarantees. Overall, this is an interesting way to combine machine learning and control.